# Building Multivariate Time-Varying Smooth Transition Correlation GARCH Models, with an Application to the Four Largest Australian Banks

**Anthony D. Hall [1], Annastiina Silvennoinen [1] and Timo Teräsvirta [2,3,\*]**

[1] National Centre for Econometric Research (NCER), Queensland University of Technology, Brisbane, QLD 4000, Australia
[2] Aarhus BSS, Aarhus University, DK-8210 Aarhus V, Denmark
[3] Center for Applied Statistics and Economics (C.A.S.E.), Humboldt-Universität zu Berlin, DE-10178 Berlin, Germany
[\*] Correspondence: tterasvirta@econ.au.dk

**Abstract:** This paper proposes a methodology for building Multivariate Time-Varying STCC–GARCH models. The novel contributions in this area are the specification tests related to the correlation component, the extension of the general model to allow for additional correlation regimes, and a detailed exposition of the systematic, improved modelling cycle required for such nonlinear models. There is an R-package that includes the steps in the modelling cycle. Simulations demonstrate the robustness of the recommended model building approach. The modelling cycle is illustrated using daily return series for Australia's four largest banks.

**Keywords:** unconditional correlation; modelling volatility; modelling correlations; multivariate autoregressive conditional heteroskedasticity

## 1. Introduction

Recently, Silvennoinen and Teräsvirta (2021) introduced a new multivariate GARCH model called the Multivariate Time-Varying Smooth Transition GARCH model (MTV model). This is a model that explicitly accounts for nonstationarities that are common in daily return series. The authors considered maximum likelihood (ML) estimation of the parameters of the model and, under suitable conditions, proved the consistency and asymptotic normality of the resulting ML estimators.

Before actually estimating an MTV model, however, the model builder has to make a number of data-driven decisions needed for specifying the parametric structure of the model. Further, the estimated structure has to be evaluated by statistical tests to reveal its potential weaknesses. Silvennoinen and Teräsvirta (2021) did not, however, discuss any model building issues, leaving them for further research. The present work is intended to fill this void.

As with many other multivariate GARCH models, the MTV model is based on the decomposition of the conditional covariance matrix Bollerslev (1990), in which the conditional covariance is decomposed to conditional variances and a conditional correlation matrix. In the MTV model, however, it is assumed that the conditional variances can be nonstationary, while a nested special case, (weak) stationarity, is a testable hypothesis.

Likewise, the correlations in this model are time-varying such that its time-varying correlation matrix nests a constant correlation matrix. Due to the parametric structure of this nonlinear correlation matrix, the constancy of correlations has to be tested (and rejected) before fitting a model with time-varying correlations.

Since, as will be explained later, these testing situations, both in the variances and the correlation matrix, are nonstandard, specification of the MTV model is an important issue in building MTV models. Furthermore, the estimated MTV model has to be evaluated before using it, from which it follows that techniques for this part of the model building process have to be examined as well.

In order to illustrate the MTV model building, we consider the Australian banking sector. This is an oligopoly dominated by four banks, commonly called the 'Big Four'. In early 2020, they represented approximately 19% of the market value of the ASX200 share index and held about 80% of the home loan market in Australia; see Figure 1. Consequently, the banking sector is a major component for many Australian superannuation and other investment funds. As to the Big Four daily returns, their volatility cannot automatically be assumed to be stationary.

Furthermore, the correlations, even when time-varying, cannot a priori be assumed to fluctuate around a constant level, which is one of the assumptions in many popular multivariate GARCH models. Applying the flexible MTV model to these return series is, therefore, an interesting exercise. An in-depth analysis of the Australian banking sector is beyond the scope of this paper; however, modelling the daily returns of the Big Four over a period of almost 30 years serves as a useful example of how our MTV model building techniques work and are applied in practice.

The modelling process is data driven, requiring user input and consists of several steps. For this reason, we developed an R-package to help users build MTV models. The version of R used is 4.1.0. The package, called `mtvgarch`, includes, among other things, the estimation routines as well as the necessary specification and evaluation tests. The code is maintained in a private GitHub repository and can be obtained upon request.

The plan of the paper is as follows. The MTV model is introduced in Section 2, followed by details of the stages and procedures related to the model building in Section 3. Model specification is considered in Section 4, estimation in Section 5 and evaluation in Section 6. Section 7 is devoted to the illustration of the complete modelling cycle on the Big Four volatilities and correlations. Our conclusions can be found in Section 8. There are also appendices containing material, such as relevant test statistics, simulation results, details of the estimation algorithm, and estimated equations.

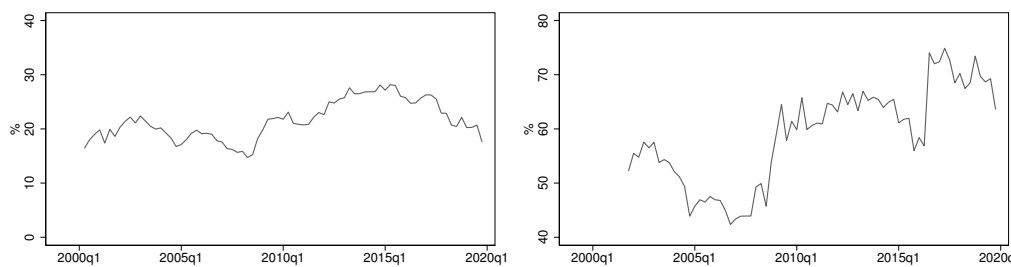

**Figure 1.** The market capitalisation of the Big Four as percentage of ASX200 (**left**) and of ASX200 Financials Index (**right**).

## 2. The MTV Model

The MTV model used in this paper belongs to the family of multivariate GARCH models introduced by Bollerslev (1990). In the original model, the conditional correlations were constant, hence, the name Constant Conditional Correlation (CCC-GARCH) model. This assumption that made the resulting model rather parsimonious was later found to be too restrictive in applications, and time-varying correlations were simultaneously introduced by Engle (2002) (dynamic conditional correlations, DCC) and Tse and Tsui (2002) (varying correlations, VC). In these models, conditional variance components are typically assumed to be stationary, and correlations are assumed, at least implicitly, to fluctuate around a constant level.

In order to consider the MTV model as defined in Silvennoinen and Teräsvirta (2021), we introduce certain notation. The observable stochastic $N \times 1$ vector $\varepsilon_t$ is decomposed in a customary fashion as

$$\varepsilon_t = H_t^{1/2} z_t = S_t D_t P_t^{1/2} \zeta_t, \tag{1}$$

where $H_t = S_t D_t P_t D_t S_t$ is an $N \times N$ covariance matrix, and $\zeta_t \sim \text{iid}(\mathbf{0}, I_N)$. We also define $z_t = P_t^{1/2} \zeta_t$, a vector of independent random variables with $\mathsf{E} z_t = \mathbf{0}$ and a positive definite deterministically varying covariance matrix $\text{cov}(z_t) = P_t$. The structure of $P_t$ will be defined later. The deterministic matrix $S_t = \text{diag}(g_{1t}^{1/2}, \ldots, g_{Nt}^{1/2})$ has positive diagonal elements for all $t$, and $D_t = \text{diag}(h_{1t}^{1/2}, \ldots, h_{Nt}^{1/2})$ contains the conditional standard deviations of the elements of $S_t^{-1} \varepsilon_t = (\varepsilon_{1t}/g_{1t}^{1/2}, \ldots, \varepsilon_{Nt}/g_{Nt}^{1/2})'$. As in Silvennoinen and Teräsvirta (2021) and earlier univariate papers, beginning with Amado and Teräsvirta (2008), and in the multivariate time-varying GARCH article by Amado and Teräsvirta (2014), the diagonal elements of $S_t^2$ are defined as follows:

$$g_{it} = g_i(t/T) = \delta_{i0} + \sum_{j=1}^{r_i} \delta_{ij} G_{ij}(t/T, \gamma_{ij}, c_{ij}), \tag{2}$$

$i = 1, \ldots, N$, where $\delta_{i0} > 0$ is a known constant, $\delta_{ij} \neq 0$, $j = 1, \ldots, r_i$, and the (generalised) logistic function

$$G_{ij}(t/T, \gamma_{ij}, c_{ij}) = (1 + \exp\{-\gamma_{ij} \prod_{k=1}^{K_{ij}} (t/T - c_{ijk})\})^{-1}, \tag{3}$$

where $\gamma_{ij} > 0$ and $c_{ij} = (c_{ij1}, \ldots, c_{ijK_{ij}})'$ such that $c_{ij1} \leq \ldots \leq c_{ijK_{ij}}$. Both $\gamma_{ij} > 0$, $c_{ij1} \leq \ldots \leq c_{ijK_{ij}}$, and $\delta_{ij} \neq 0$, $j = 1, \ldots, r_i$ are identification restrictions. Assuming $\delta_{i0}$ in (2) is known is another one. Furthermore, to prevent exchangeability of the components in (2), restrictions are needed on $c_{ij}$. As an example, if $K_{ij} = 1$ for $j = 1, \ldots, r_i$, one can assume (for instance) that $c_{i11} < \ldots < c_{ir_1 1}$.

As discussed in earlier papers, the idea of $g_{it}$ is to normalise or rescale the observations. Left-multiplying (1) by $S_t^{-1}$ yields

$$\phi_t = S_t^{-1} \varepsilon_t = D_t z_t,$$

where each element of $\phi_t$ is assumed to have a standard weakly stationary GARCH representation. In our work, the conditional variances have a GARCH or GJR-GARCH(1,1) structure; see Glosten et al. (1993) for the latter:

$$h_{it} = \alpha_{i0} + \alpha_{i1} \phi_{i,t-1}^2 + \kappa_{i1} I(\phi_{i,t-1} < 0) \phi_{i,t-1}^2 + \beta_{i1} h_{i,t-1}, \tag{4}$$

where $I(A)$ is an indicator function: $I(A) = 1$ when $A$ occurs, zero otherwise. A higher-order structure is possible, although there do not seem to exist applications of the GJR-GARCH model of order greater than one.

The conditional covariance matrix $\mathsf{E}\{\phi_t \phi_t' | \mathcal{F}_{t-1}\} = D_t P_t D_t$. In order to describe the correlation structure, we employ the Double Smooth Transition Conditional Correlation (DSTCC) model by Silvennoinen and Teräsvirta (2009). In that model, assuming that the transition variable is $t/T$ throughout, the time-varying correlation matrix $P_t$ is defined as

$$P_t = (1 - G_2(t/T, \gamma_2, c_2))\{(1 - G_1(t/T, \gamma_1, c_1))P_{(11)} + G_1(t/T, \gamma_1, c_1)P_{(21)}\}$$
$$+ G_2(t/T, \gamma_2, c_2)\{(1 - G_1(t/T, \gamma_1, c_1))P_{(12)} + G_1(t/T, \gamma_1, c_1)P_{(22)}\}, \tag{5}$$

where $P_{(ij)}$, $i, j = 1, 2$, are four positive definite correlation matrices not equal to each other, and

$$G_i(t/T, \gamma_i, c_i) = (1 + \exp\{-\gamma_i \prod_{k=1}^{K_i} (t/T - c_{ik})\})^{-1}, \quad \gamma_i > 0 \tag{6}$$

where $c_i = (c_{i1}, \ldots, c_{iK_i})$, $c_{i1} < \ldots < c_{iK_i}$, $i = 1, 2$. This variant of the DSTCC model is called the Time-Varying Correlation (TVC) model to emphasise its deterministic rather than stochastic nature—hence, removing the term 'Conditional' from its name. For the Big Four application, we simplify the definition (5) slightly by assuming $\boldsymbol{P}_{(12)} = \boldsymbol{P}_{(22)}$, and therefore (5) becomes

$$\boldsymbol{P}_t = (1 - G_2(t/T, \gamma_2, \boldsymbol{c}_2))\{(1 - G_1(t/T, \gamma_1, \boldsymbol{c}_1))\boldsymbol{P}_{(1)} + G_1(t/T, \gamma_1, \boldsymbol{c}_1)\boldsymbol{P}_{(2)}\}$$
$$+ G_2(t/T, \gamma_2, \boldsymbol{c}_2)\boldsymbol{P}_{(3)}, \tag{7}$$

where re-indexing the matrices highlights the interpretation that there are two transitions over time. One is from $\boldsymbol{P}_{(1)}$ to $\boldsymbol{P}_{(2)}$, and the other one is from a convex combination of these two to $\boldsymbol{P}_{(3)}$. Since $\boldsymbol{P}_{(1)}$, $\boldsymbol{P}_{(2)}$, and $\boldsymbol{P}_{(3)}$ are positive definite, $\boldsymbol{P}_t$ is positive definite as a convex combination of the three matrices. This simplified version of the TVC model is especially useful when modelling correlations that shift from one state to the next as a function of time. To that end, the obvious extension to $n$ such transitions is best expressed as a recursion

$$\boldsymbol{P}_t^{(0)} = \boldsymbol{P}_{(1)}$$
$$\boldsymbol{P}_t^{(n)} = (1 - G_n(t/T, \gamma_n, \boldsymbol{c}_n))\boldsymbol{P}_t^{(n-1)} + G_n(t/T, \gamma_n, \boldsymbol{c}_n)\boldsymbol{P}_{(n+1)}. \tag{8}$$

When $G_2(t/T, \gamma_2, \boldsymbol{c}_2) \equiv 1$ and $N = 2$, (5) and (7) collapse into the smooth transition correlation GARCH model by Berben and Jansen (2005) or, if the transition variable in $G_1$ is stochastic and $N \geq 2$, into the smooth transition conditional correlation GARCH model of Silvennoinen and Teräsvirta (2005, 2015). An MTV-Conditional Correlation GARCH model with GARCH equations similar to the ones here but differently defined stochastic $\boldsymbol{P}_t$ was discussed in Amado and Teräsvirta (2014). It may be noted that Feng (2006) introduced another multivariate Conditional Correlation type GARCH model with deterministically varying correlations. In this model, the variation is described nonparametrically, and the model can be viewed as a generalisation of the univariate model in Feng (2004).

## 3. The Three Stages of Model Building

The MTV model is rather general and nests many models. To take one example, fitting an MTV model when a nested CCC-GARCH model generates the data leads to inconsistent parameter estimates. For this reason, building adequate MTV models requires care, and a systematic approach is necessary. Selecting a candidate from this family of models is a data-driven process, and statistical inference has to be used to obtain an acceptable model such that it passes the available misspecification tests.

In this work, we follow the classical approach to model building advocated by Box and Jenkins (1970) and later applied to nonlinear models of the conditional mean; see, for example, Teräsvirta et al. (2010, Ch. 16). It has also been applied to building single-equation MTV-GARCH models; see Amado and Teräsvirta (2017) and Amado et al. (2017). The idea is to first specify the model (select a member from the family of MTV models) and, once this has been done, to estimate its parameters. At the evaluation stage, the estimated model is subjected to a battery of misspecification tests. These three stages, specification, estimation, and evaluation, will be considered in the next three sections. The emphasis will be on specification and evaluation as maximum likelihood estimation of the parameters of the MTV model was already considered in Silvennoinen and Teräsvirta (2021).

## 4. Specification of the MTV Model
### 4.1. Specification of the Univariate Variance Equations

Specification of the MTV model is begun by specifying the univariate volatility equations. This was first discussed in Amado and Teräsvirta (2017). The idea is to begin with a GARCH(1,1) model by Bollerslev (1986) or the GJR-GARCH model by Glosten et al. (1993)

and to test the hypothesis that the multiplicative deterministic component is constant. The single-equation MTV-GARCH model has the following form:

$$\varepsilon_{it} = z_{it} h_{it}^{1/2} g_{it}^{1/2}, \tag{9}$$

where $z_{it} \sim \text{iid}(0,1)$, the conditional variance $h_{it}$ is defined as in (4) with $\phi_{it} = \varepsilon_{it}/g_{it}^{1/2}$, and the deterministic positive-valued function $g_{it} = g_i(t/T)$ is defined as in (2) and (3).

Positivity of (2) imposes the following restrictions on $\delta_{ij}$, $j = 1, \ldots, r_i$:

$$\delta_{i0} + \sum_{j=1}^{r_i} \delta_{ij} G_{ij}(r, \gamma_{ij}, c_{ij}) > 0$$

for all $r \in [0,1]$.

Typically in applications, $K_{ij} = 1, 2$ in (3). There are two specification issues, determining $r_i$ and choosing $K_{ij}$, $j = 1, \ldots, r_i$. It is possible that $g_i(t/T) = \delta_0 > 0$—that is, $g_i(t/T)$ is a positive constant. In this case, the MTV-GARCH model collapses into a standard GARCH or GJR-GARCH equation.

Amado and Teräsvirta (2017) solved the problem of choosing $r_i$ by first estimating the GARCH model and testing the hypothesis of a constant $g_i(t/T)$ against the alternative $r_i = 1$ in (2) thereafter using a Lagrange multiplier type test. The test can be viewed as a misspecification test of the estimated GARCH model. If the null hypothesis is rejected, an MTV-GARCH model with a single transition is estimated, and the hypothesis $r_i = 1$ is tested against $r_i = 2$. Sequential testing continues until the first non-rejection of the null hypothesis.

The number of transitions is determined in this order because of an identification problem: the model with $r_i + 1$ transitions is not identified if the true number of transitions is $r_i$. The shape of the logistic function, controlled by the parameter $K_{ij}$, can be determined using the sequence of tests familiar from the specification of smooth transition autoregressive (STAR) models; see Teräsvirta (1994) or Teräsvirta et al. (2010, Ch. 16). Details can be found in Amado and Teräsvirta (2017).

More recently, Silvennoinen and Teräsvirta (2016) considered testing the constancy of $g_i(t/T)$ before estimating the GARCH model—that is, assuming $h_{it} = 1$ in (9). The details are laid out in Appendix A.1. This implies that the size of the test is distorted because conditional heteroskedasticity is ignored, so it has to be adjusted by simulation. It turns out that, by doing this, the power of the size-adjusted test considerably improves compared to the case where the test is a standard misspecification test. Reasons for this improvement are discussed in Silvennoinen and Teräsvirta (2016).

A major difficulty with this approach is that, while in simulations, the parameters of the conditional variance component $h_{it}$ under the null hypothesis are known—in practice, this is not the case. The underlying 'null' GARCH process has to be generated artificially. In so doing, special attention is to be placed on the persistence of the (GJR-)GARCH process, measured by $\alpha_{i1} + \kappa_{i1}/2 + \beta_{i1}$ in (4) when $g_{it} \equiv 1$. In fact, the asymmetry parameter has no practical importance for the purpose of calibrating the test statistic distribution, and it is therefore sufficient to restrict attention to the standard GARCH process. Other features, such as implied kurtosis or relative sizes of $\alpha$ and $\beta$ corresponding to a particular level of persistence only have a negligible effect on the performance of the test.

A practical problem is that it is not possible to estimate this measure of persistence when the null hypothesis does not hold—that is, when $g_{it}$ is not constant over time. How this difficulty is handled has an effect on the power of the test. We study two approaches that are discussed more in detail in Appendix B.1. The first one consists of visually identifying a period of time where there appears to be no change in the overall level of baseline volatility. A standard GARCH(1,1) is estimated over this subperiod. The second approach is to use rolling window variance targeting. This means that the intercept in the

GARCH equation is time-varying, and its value at each point in time is calculated such that it matches the unconditional variance obtained from a window around that point in time.

Simulations discussed in Appendix B.1 experiment with the choice of window size. Both of these methods provide GARCH parameter and persistence estimates that are used for calibrating the null distribution of the test statistic and calculating *p*-values.

### 4.2. Specification of Time-Varying Correlations

After the MTV-GARCH equations have been specified and estimated assuming the errors are uncorrelated, the next step is to specify the time-varying correlation structure. This is done by sequential testing. First, the constancy of correlation tested against the model with a single transition, i.e., $G_2(t/T, \gamma_2, c_2) \equiv 1$ in (5). The null hypothesis is that the model is a MTV-Constant Correlation GARCH model as in Bollerslev (1990), except that the GARCH equations are MTV-GARCH equations. If this model is rejected, the one-transition model is estimated and tested against (5) or (7). If this specification is also rejected, the alternative with two transitions is estimated. This is repeated until no further evidence for time-variation in the correlations is detected.

As discussed in Silvennoinen and Teräsvirta (2005, 2015), the MTV model with one transition is only identified under the alternative, which invalidates the standard asymptotic inference. The identification problem can be circumvented by approximating the transition function (6) by its Taylor expansion around the null hypothesis, $H_0$: $\gamma_1 = 0$. The form of the expansion depends on the order of the exponent in (6).

The test can be constructed along the lines presented in the appendix of Silvennoinen and Teräsvirta (2005).[1] See also Silvennoinen and Teräsvirta (2021). To derive the test statistic, consider the first-order Taylor expansion of (6) around $\gamma_1 = 0$ assuming $K_1 = 2$. It has the following form:

$$
\begin{aligned}
G_1(t/T, \gamma_1, c_1) &= (1 + \exp\{-\gamma_1 \prod_{k=1}^{K_1}(t/T - c_{1k})\})^{-1} \\
&= \frac{1}{2} + \frac{1}{4}(t/T - c_{11})(t/T - c_{12})\gamma_1 + R_2(t/T; \gamma_1),
\end{aligned}
\tag{10}
$$

where $R_2(t/T; \gamma_1)$ is the remainder. Using (10), (5) becomes

$$
\begin{aligned}
\boldsymbol{P}_t &= (\boldsymbol{P}_{(1)} - \boldsymbol{P}_{(2)})(\frac{1}{2} + \frac{\gamma_1 c_{11} c_{12}}{4}) + \boldsymbol{P}_{(2)} - (t/T)(\boldsymbol{P}_{(1)} - \boldsymbol{P}_{(2)})\frac{\gamma_1(c_{11} + c_{12})}{4} \\
&\quad + (t/T)^2(\boldsymbol{P}_{(1)} - \boldsymbol{P}_{(2)})\frac{\gamma_1}{4} + (\boldsymbol{P}_{(1)} - \boldsymbol{P}_{(2)})R_2(t/T; \gamma_1) \\
&= \boldsymbol{P}_{(A0)} + (t/T)\boldsymbol{P}_{(A1)} + (t/T)^2\boldsymbol{P}_{(A2)} + (\boldsymbol{P}_{(1)} - \boldsymbol{P}_{(2)})R_2(t/T; \gamma_1),
\end{aligned}
$$

where $\boldsymbol{P}_{(A0)} = (\boldsymbol{P}_{(1)} - \boldsymbol{P}_{(2)})(1/2 + \gamma_1 c_{11} c_{22}/4) + \boldsymbol{P}_{(2)}$, $\boldsymbol{P}_{(A1)} = -(\boldsymbol{P}_{(1)} - \boldsymbol{P}_{(2)})\gamma_1(c_{11} + c_{12})/4$, $\boldsymbol{P}_{(A2)} = (\boldsymbol{P}_{(1)} - \boldsymbol{P}_{(2)})\gamma_1/4$, and $\boldsymbol{P}_{(1)} \neq \boldsymbol{P}_{(2)}$. The main diagonals of $\boldsymbol{P}_{(A1)}$ and $\boldsymbol{P}_{(A2)}$ consist of zeroes. Setting $\boldsymbol{\rho}_A = (\boldsymbol{\rho}'_{A0}, \boldsymbol{\rho}'_{A1}, \boldsymbol{\rho}'_{A2})'$, where $\boldsymbol{\rho}_{Ai} = \text{vecl}(\boldsymbol{P}_{(Ai)})$, $i = 0, 1, 2$, the new null hypothesis is $H_0$: $\boldsymbol{\rho}_{A1} = \boldsymbol{\rho}_{A2} = \boldsymbol{0}_{N(N-1)/2}$.[2]

Note that a simpler version of the test assumes $K_i = 1$ and yields a similar approximation although without the term $(t/T)^2 \boldsymbol{P}_{(A2)}$. The new null in this case is $H_0$: $\boldsymbol{\rho}_{A1} = \boldsymbol{0}_{N(N-1)/2}$. This version of the test is more powerful than the former in the case that time-variation in the correlations is monotonic. However, and especially with longer time horizons, this may not always be the case, and the square term of the expansion is able to capture at least some nonmonotonic changes.

The details of the ensuing LM-type test statistic for the test of constant correlations is presented in Appendix A.3, and the test for an additional transition in correlations is laid out in Appendix A.4.

## 5. Estimation of the MTV Model

After specifying the deterministic components of the model, both in GARCH equations and correlations, one can estimate the complete model with conditional heteroskedasticity included. The log-likelihood of the MTV-STCC-GARCH model has the form

$$
\ln f(\boldsymbol{\zeta}_t | \boldsymbol{\theta}) \propto - (1/2) \sum_{i=1}^{N} \ln g_{it}(\boldsymbol{\theta}_{gi}) - (1/2) \sum_{i=1}^{N} \ln h_{it}(\boldsymbol{\theta}_{hi}) - (1/2) \ln |\boldsymbol{P}_t(\boldsymbol{\theta}_P)|
$$
$$
- (1/2) \boldsymbol{\varepsilon}_t' \{ \boldsymbol{S}_t(\boldsymbol{\theta}_g) \boldsymbol{D}_t(\boldsymbol{\theta}_g, \boldsymbol{\theta}_h) \boldsymbol{P}_t(\boldsymbol{\theta}_P) \boldsymbol{D}_t(\boldsymbol{\theta}_g, \boldsymbol{\theta}_h) \boldsymbol{S}_t(\boldsymbol{\theta}_g) \}^{-1} \boldsymbol{\varepsilon}_t, \tag{11}
$$

where the full parameter vector $\boldsymbol{\theta} = (\boldsymbol{\theta}_h', \boldsymbol{\theta}_g', \boldsymbol{\theta}_P')'$ is partitioned according to the relevant functions: the conditional variance in (4) $\boldsymbol{\theta}_h = (\boldsymbol{\theta}_{h1}', \ldots, \boldsymbol{\theta}_{hN}')'$ with $\boldsymbol{\theta}_{hi} = (\alpha_{i0}, \alpha_{i1}, \kappa_{i1}, \beta_{i1})'$, $i = 1, \ldots, N$; the deterministic variance component in (2) and (3) $\boldsymbol{\theta}_g = (\boldsymbol{\theta}_{g1}', \ldots, \boldsymbol{\theta}_{gN}')'$ with $\boldsymbol{\theta}_{gi} = (\delta_{i1}, \gamma_{i1}, \boldsymbol{c}_{i1}, \ldots, \delta_{ir_i}, \gamma_{ir_i}, \boldsymbol{c}_{ir_i})'$, $i = 1, \ldots, N$; and correlations $\boldsymbol{\theta}_P = (\text{vecl} \boldsymbol{P}_{(1)}, \ldots, \gamma_1, \ldots, \boldsymbol{c}_1, \ldots)'$, where the number of matrices as well as transition functions (6) are determined by the choice of the model, (5), (7), or (8).

We make the following assumptions; see Silvennoinen and Teräsvirta (2021):

AN1. In (4), $\alpha_{i0} > 0$, either $\alpha_{i1} > 0$ and $\alpha_{i1} + \kappa_{i1} \geq 0$ or $\alpha_{i1} \geq 0$ and $\alpha_{i1} + \kappa_{i1} > 0$, $\beta_{i1} \geq 0$, and $\alpha_{i1} + \kappa_{i1}/2 + \beta_{i1} < 1$ for $i = 1, \ldots, N$.

AN2. The parameter subspaces $\{\alpha_{i0} \times \alpha_{i1} \times \kappa_{i1} \times \beta_{i1}\}$, $i = 1, \ldots, N$, are compact, the whole space $\Theta_h$ is compact, and the true parameter value $\boldsymbol{\theta}_h^0$ is an interior point of $\Theta_h$.

AN3. $\boldsymbol{\zeta}_t \sim \text{iid} N(\boldsymbol{0}, \boldsymbol{I}_N)$.

AN1 is the necessary and sufficient weak stationarity condition for the $i$th first-order GJR-GARCH equation. Assumption AN2 is a standard regularity condition required for proving the asymptotic normality of maximum likelihood estimators of $\boldsymbol{\theta}_{hi}$, $i = 1, \ldots, N$. AN3 (normality) is a strong condition; however, it is needed here for the proofs to go through; see Silvennoinen and Teräsvirta (2021). These assumptions are sufficient for the maximum likelihood estimators of the GARCH parameters in single-equation GARCH models to be consistent and asymptotically normal.

The parameters are estimated in turn: first estimate $\boldsymbol{\theta}_{gi}$ to obtain starting-values for the joint estimation of $\boldsymbol{\theta}_g$ and $\boldsymbol{\theta}_P$. This is done assuming $h_{it}(\boldsymbol{\theta}_{hi}) \equiv 1$, $i = 1, \ldots, N$. Amado and Teräsvirta (2013) showed in the single-equation GJR-GARCH case that, under regularity conditions, the maximum likelihood estimator of $\boldsymbol{\theta}_{gi}$ is consistent and asymptotically normal. Silvennoinen and Teräsvirta (2021) generalised this result to MTV models. That means that joint estimation of $\boldsymbol{\theta}_g$ and $\boldsymbol{\theta}_P$ by maximum likelihood produces consistent estimates of these parameter vectors.

If $\widehat{\boldsymbol{\theta}}_g$ and $\widehat{\boldsymbol{\theta}}_P$ are consistent and Assumptions AN1, AN2, and AN3 hold, then, by Theorem 3.3 of Song et al. (2005), the maximum likelihood estimator of $\boldsymbol{\theta}_h$ is consistent and asymptotically normal. After estimating $\boldsymbol{\theta}_h$, the parameter vectors $\boldsymbol{\theta}_g$ and $\boldsymbol{\theta}_P$ are re-estimated. Iteration continues until convergence. Song et al. (2005) showed that the final maximum likelihood estimator of $\boldsymbol{\theta}$ is consistent and asymptotically normal. A more detailed description of the maximisation by parts applied to the present situation can be found in Appendix C; see also Silvennoinen and Teräsvirta (2021).

## 6. Evaluation of the MTV Model

Once the model has been specified and estimated, it has to be evaluated in order to find potential misspecifications. The tests in Section 4.1 were used to guide the choice of the functional form of the deterministic component, and a rejection of the null was seen as evidence of the current model still lacking in its specification. In that sense, the tests in Section 4.1 are seen as both specification and evaluation tests. It is worth reiterating that these specification tests were constructed at the stage when the GARCH part was not yet specified—that is, $h_t = 1$ in (4). However, when the deterministic part passes these tests, and an MTV-GARCH equation is subsequently estimated, there is room for additional checks in terms of model misspecification, beyond the presently final model specification.

The tests in Amado and Teräsvirta (2017) are available for this purpose. They fall into three categories. In the first one, the deterministic component is additively misspecified. In the context of the current MTV-GARCH model, the relevant case is a test for yet another transition in (2). The second test assesses the GARCH equation for additive misspecification. The concern here is the validity of the maximum lags $p$ or $q$. The final test is the 'test of no remaining ARCH', which is based on the idea of a sufficiently well-specified model managing to clear any autocorrelation from the squared standardised residuals. The test that suits each of these situations (or its robustified version to avoid the assumption of normality) is conveniently performed following a set of steps outlined in Appendix A.2.

It is worth stating that the tests here are applied to $\hat{P}_t^{-1/2}\varepsilon_t$, one series at a time. Efforts towards completing the tests in the complete $N$-variate system simultaneously would open up a vast number of permutations of various misspecification options. To manage the task, the recommendation is to focus on the univariate specifications one at a time, even with the acknowledgment of some potential for deviating from the asymptotically exact results. Simulations in Section Appendix B.2 indicate that applying the tests on the pre-filtered data has very little impact on the distributions of the test statistics. While the standard form of the misspecification tests suffers from minor oversizing, this is mostly corrected when using the robust version of the test.

The test for an additional transition in the correlations in Section 4.2 may also be used as an evaluation test. It is based on the completely specified univariate and correlation components, and therefore its role as a misspecification test of a complete model is justified. The number of degrees of freedom in this test quickly becomes large with increasing $N$. One way of restricting this growth would be to assume that, under the alternative, only the eigenvalues of the correlation matrix are changing over time. The alternative would be a correlation matrix only if all correlations were identical (see Engle and Kelly (2012)); however, an LM test can nevertheless be built on this assumption.

Write the correlation matrix as $P_t = Q_t\Lambda_t Q_t'$, where $P_t$ is defined as in (5), $\Lambda_t$ is the diagonal matrix of eigenvalues and $Q_t$ contains the corresponding eigenvectors. Simplify this by assuming $Q_t = Q$ and approximate $\Lambda_t$ by $\Psi_t = \sum_{k=0}^K \Psi_k (t/T)^k$. Under the null hypothesis, $K = 0$. The resulting test statistic is derived, and its small-sample properties are studied in Kang et al. (2022).

## 7. Big Four Results

### 7.1. Main Features of the Australian Banking Sector 1990–2020

In order to provide some background for our empirical results, we shall now draw attention to a number of interesting features of the Australian banking sector between the years 1990 and 2020. In 1990, the Australian government adopted an intervention policy called 'six pillars'. It covered the four biggest Australian banks commonly referred to as the 'Big Four', the Commonwealth Bank of Australia (CBA), Westpac Banking Corporation (WBC), National Australia Bank (NAB) and Australia and New Zealand Banking Group (ANZ), listed in descending order of market share, as well as two insurers (AMP Limited and National Mutual).

This policy stated that further mergers of these institutions would not be accepted. The basic idea was to ensure a competitive banking market. In 1997, the policy became 'four pillars' as the insurers were left outside the arrangement. Since its establishment, it has mostly enjoyed the support of the two main political parties, and the proponents of the policy have argued that it has contributed to the stability and strength of the Australian financial sector. The government also had sympathetic policy settings, which allowed the banks to recapitalize in the 1990s and 2000s.

During this period, there were financial losses at Westpac (a $1.6 billion loss in 1992 and close to insolvency), ANZ (poorly executed international expansions) and subsequently with NAB (purchase of the US mortgage originator and servicer Homeside led to $2.2 billion in losses 2002). Although the Big Four were not allowed to merge with each other, larger financial concentration due to mergers with other financial institutions was seen as accept-

able: in 2008, Westpac and CBA acquired St. George and BankWest and then the fifth and the sixth largest Australian banks, respectively. The impact can be seen in the right panel of Figure 1 as the Big Four's share of the ASX200 Financials Index drastically increased.

The recent history of the pillars policy coincides with a few major incidents and changes. These include not only the dot-com boom in the late 1990s and early 2000s and the global financial crisis (GFC) nearly ten years later but also events that have had more localised impacts, such as a number of of regulatory changes (Basel guidelines), the most recent mining boom that started around 2005 and was interrupted by the GFC, and technology-driven market disruptions (non-bank lenders and payment providers). Since the GFC, the banks have enjoyed substantial government support, including a deposit guarantee and, as already noted, have come to dominate the home mortgage market with an 80% market share.

During the first decade of the millennium, a few of the above-mentioned events have positioned the Big Four in an increasingly competitive environment. The stagnation of the housing market and the removal of barriers to changing mortgage providers may also have contributed towards this trend; however, the most notable event was the announcement in 2003 that Basel II was to be implemented in Australia by end of 2007. The idea behind the updated accord was to level inequalities amongst the internationally active banks and to set expectations regarding capital adequacy requirements. The Australian Prudential Regulation Authority, in charge of overseeing the uptake of the accord, worked extensively with numerous Authorised Deposit-Taking Institutions, the industry and other relevant bodies during 2005 to 2007, aiming at ensuring that the adoption of Basel II included all relevant aspects of the implementation process, its goals, and impacts.

Fear of being subjected to a competitive disadvantage relative to their international counterparts, both within international and domestic operations, coupled with an opportunity for a reduced regulatory capital, incentivised the banks to signal early their preference to conform with the accord.[3] As a result, Australia was amongst the first nations to have fully implemented the framework on 1 January 2008.

### 7.2. Modelling the Error Variances

The daily return series for the Big Four used in this paper extend from 2 January 1992 to 31 January 2020. As suggested in the Introduction, these series may not be adequately described by a weakly stationary GARCH or GJR-GARCH model. From the plots in Figure 2, it is seen that the amplitude of clusters varies for all four banks, in particular during and after the financial crisis beginning in 2008. The crisis was preceded by a rather tranquil period between 2003 and 2008. This variation also shows in the autocorrelation functions of the squared returns in Figure 3. In all four cases, the autocorrelations decay very slowly as a function of the lag length, which suggests nonstationarity.

For this reason, modelling the returns has to be initiated by testing the stationarity hypothesis. As discussed in Section 4.1, the slow moving 'baseline' volatility is specified first, followed by the inclusion of the GJR-GARCH component. The test statistic from Appendix A.1 is calibrated using methods described in Appendix B.1. In the first one, based on choosing a period during which the amplitude of clusters seems constant, the selected period extends from November 2003 to October 2007. In the other, where a rolling window is moved over the observation period, the window size chosen by simulations is 400, as outlined in Appendix B.1.

To enhance the performance of the test, the entire sample of over 7000 observations is broken into subsections. Once one transition is found and estimated, the test is applied to both before and after this transition to determine if there is another transition on either side. The process is continued until the null of no transition is not rejected. After the deterministic component has been specified and its parameters tentatively estimated, the GJR-GARCH equations are estimated together with the time-varying component to form a complete TV-GJR-GARCH model. The estimated equations are then checked for signs of misspecification using the tests from Amado and Teräsvirta (2017).

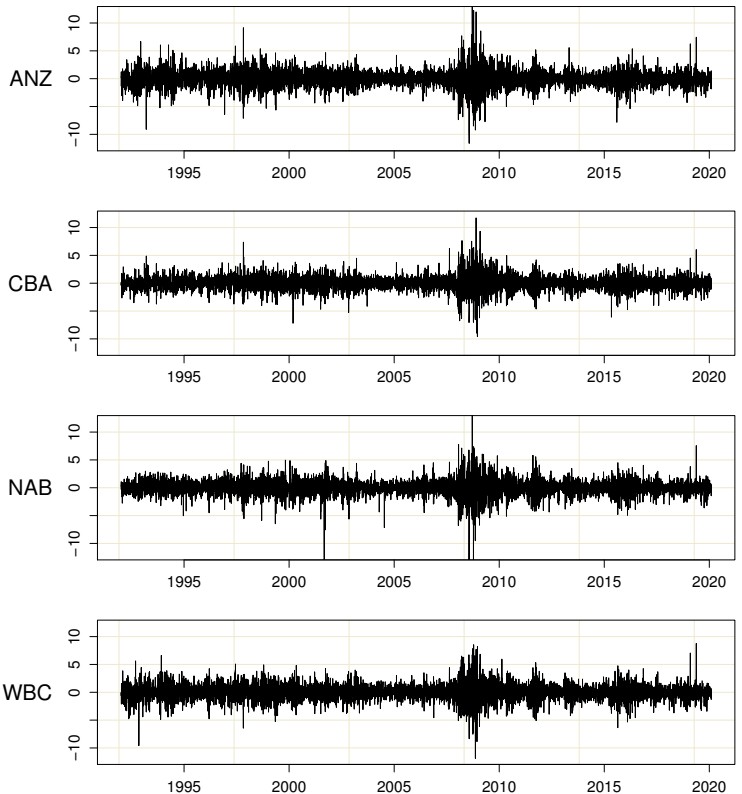

**Figure 2.** Daily returns of the Big Four, from 2 January 1992 to 31 January 2020.

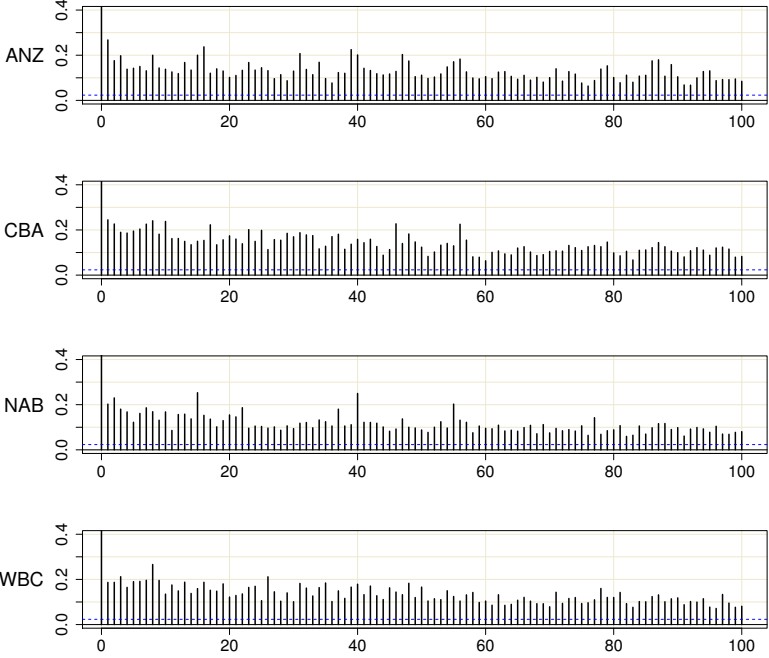

**Figure 3.** The first 100 autocorrelations of squared returns.

Estimated TV-GJR-GARCH equations results appear in Table 1, and the deterministic components are in Appendix D. For comparison, the GJR-GARCH equations without the deterministic component are also reported in this table. In all four cases, the persistence strongly decreases after rescaling the returns with the TV component.

This is also indirectly obvious from the autocorrelations in Figure 4 that are considerably smaller and decay faster than the ones in Figure 3. The main cause for this decrease lies in the coefficient of the lagged conditional variance whose estimate shrinks in the process. Estimates of the asymmetry parameter $\kappa_{i1}$ slightly increase, and thus asymmetry becomes more pronounced when nonstationarity is properly modelled. Table 1 also contains the kurtosis estimates for the two GJR-GARCH processes, obtained using definitions in He and Teräsvirta (1999). It is seen that, in all four cases, rescaling lowers the kurtosis to values close to three.

**Table 1.** Univariate estimation results for the four banks. GJR is the GJR-GARCH(1,1) equation, and TV-GJR is the TV-GJR-GARCH equation; standard errors in parentheses.

|  |  | $\alpha_{i0}$ | $\alpha_{i1}$ | $\kappa_{i1}$ | $\beta_{i1}$ | Persistence | Kurtosis |
|---|---|---|---|---|---|---|---|
| ANZ | GJR | 0.020 (0.005) | 0.039 (0.007) | 0.044 (0.008) | 0.929 (0.008) | 0.991 | 3.76 |
|  | TV-GJR | 0.111 (0.016) | 0.015 (0.005) | 0.046 (0.007) | 0.792 (0.027) | 0.831 | 3.02 |
| CBA | GJR | 0.035 (0.005) | 0.060 (0.008) | 0.063 (0.011) | 0.886 (0.010) | 0.977 | 3.66 |
|  | TV-GJR | 0.107 (0.014) | 0.021 (0.006) | 0.065 (0.010) | 0.813 (0.020) | 0.867 | 3.06 |
| NAB | GJR | 0.065 (0.009) | 0.077 (0.010) | 0.075 (0.014) | 0.850 (0.014) | 0.964 | 3.68 |
|  | TV-GJR | 0.152 (0.019) | 0.021 (0.006) | 0.058 (0.009) | 0.731 (0.030) | 0.780 | 3.03 |
| WBC | GJR | 0.031 (0.006) | 0.045 (0.007) | 0.058 (0.010) | 0.910 (0.009) | 0.985 | 3.70 |
|  | TV-GJR | 0.079 (0.011) | 0.015 (0.004) | 0.041 (0.006) | 0.829 (0.020) | 0.864 | 3.02 |

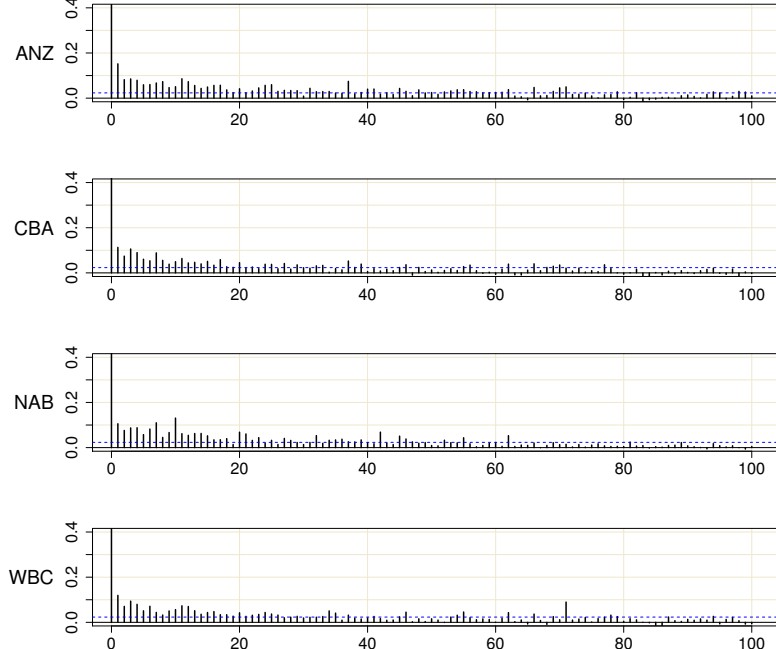

**Figure 4.** The first 100 autocorrelations of squared standardised returns $\varepsilon_{it}^2 / \widehat{g}_{it}$.

Figure 5 contains the estimated transitions. There are two conspicuous features in these graphs. One is the downward shift around 2004, which, for WBC, is a long and rather smooth decline. This coincides with the local events discussed in the previous Section. The other is that, for all four banks, the deterministic component remains higher after 2010 than it was before 2008.

For WBC, the deterministic component slowly but steadily declines after the crisis, whereas, for the three others, it remains constant. This can also be seen from the estimated equations behind the figures reported in Appendix D.

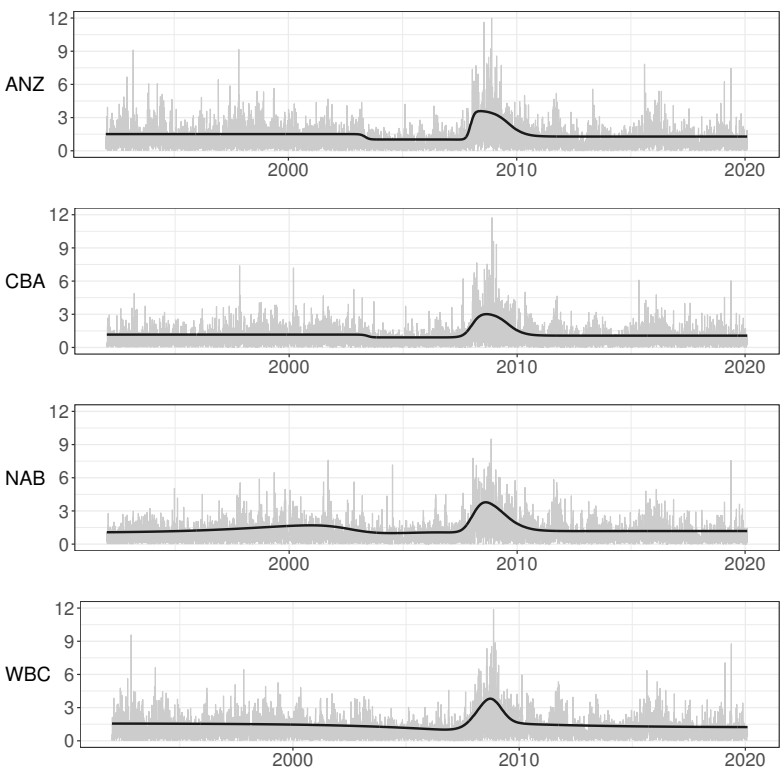

**Figure 5.** Estimated multiplicative component $\widehat{g}_{it}^{1/2}$ (solid curve) and the absolute returns $|\varepsilon_{it}|$ (grey area).

Effects of the deterministic component $g_{it}$ on the GARCH equations also become obvious by comparing the conditional variances from the GJR-GARCH equations in Figure 6 with the TV-GJR-GARCH ones in Figure 7. Clearly, for all four banks, the nonstationarity around 2008–2010 in the former figure is no longer visible in the latter.

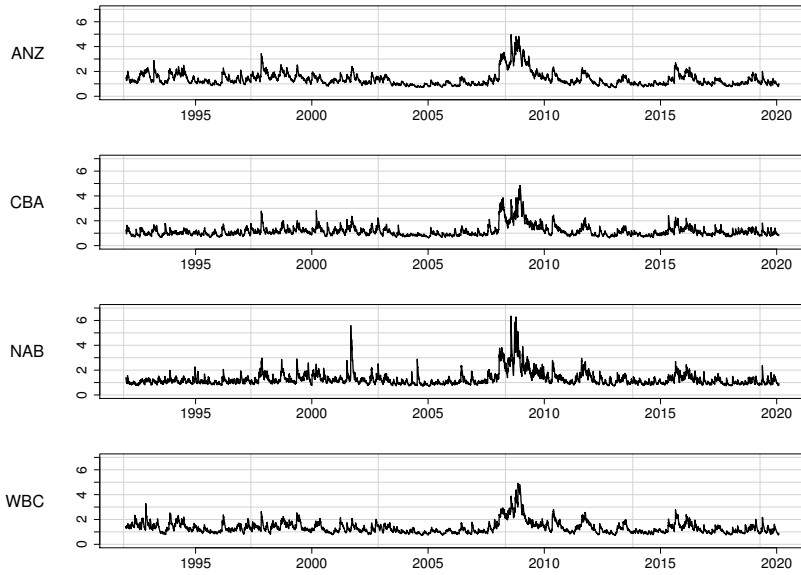

**Figure 6.** Estimated conditional variance $\widehat{h}_{it}$ from the GJR-GARCH model.

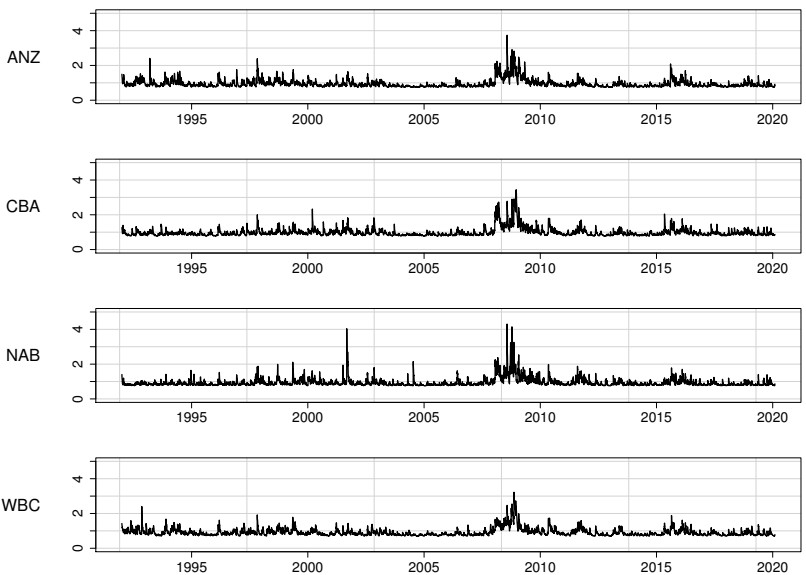

**Figure 7.** Estimated conditional variance $\widehat{h}_{it}$ from the TV-GJR-GARCH model.

### 7.3. Modelling the Error Correlations

The stability of correlations over time, as discussed in Section 4.2, is tested using the test statistic (A5) in Appendix A.3. The *p*-value of the test is very close to zero, and thus the null hypothesis is rejected. An TVC model is then estimated with a single monotonic time transition. The adequacy of this specification is tested with the test for an additional transition.

The resulting *p*-value is 0.467; therefore, the single time-transition is deemed sufficient. The estimation results of the TVC component of the model are presented in Table 2 (see also Figure 8) indicate that the correlations between the standardised residuals are stable, around 0.49–0.60 depending on the bank, from 1992 until the mid-to-late 2006. At that point, the correlations begin their steady increase to the range of 0.78–0.83, which they reach by early 2008. The final correlations are not only large but also remarkably similar.

**Table 2.** Estimation results for the four banks' time-varying correlations. A total of 90% of the estimated transition is between the dates 18 October 2006 and 28 February 2008. The centre point of the location corresponds to 28 June 2007 with ± two standard error ranges of 11 May–13 August 2007.

| | $P_{(1)}$ | | | | | $P_{(2)}$ | | |
|---|---|---|---|---|---|---|---|---|
| | ANZ | CBA | NAB | | | ANZ | CBA | NAB |
| CBA | 0.485 | | | | CBA | 0.782 | | |
| | (0.011) | | | | | (0.006) | | |
| NAB | 0.503 | 0.525 | | | NAB | 0.808 | 0.787 | |
| | (0.010) | (0.010) | | | | (0.005) | (0.005) | |
| WBC | 0.606 | 0.500 | 0.492 | | WBC | 0.830 | 0.818 | 0.814 |
| | (0.009) | (0.011) | (0.011) | | | (0.004) | (0.005) | (0.005) |

| Transition parameters: | | $c$ | $\eta$ |
|---|---|---|---|
| | | 0.552 | 5.020 |
| | | (0.002) | (0.162) |

Note: $\eta = \ln \gamma$; see Appendix A.2.

In addition to considering the complete Big Four system, we repeated the analysis using bivariate models. The outcome was similar: a single transition was sufficient for all pairs. Furthermore, inspection of the pairwise correlation estimates in Table 3 reveals strong similarity between the four-variate and bivariate estimates. This is illustrated in Figure 9. It should be noted that the shift in the correlation of the ANZ–NAB pair is estimated as a step function.

This happens when the speed of transition increases without a bound due to the likelihood function effectively becoming 'flat' with respect to that particular parameter. The solution is to fix the speed to a large value and proceed with the estimation of the remaining parameters.

**Table 3.** Estimation results for the four banks' time-varying bivariate correlations.

| | $P_{(1)}$ | | | | $P_{(2)}$ | | |
|---|---|---|---|---|---|---|---|
| | ANZ | CBA | NAB | | ANZ | CBA | NAB |
| CBA | 0.484 (0.011) | | | CBA | 0.784 (0.006) | | |
| NAB | 0.510 (0.010) | 0.518 (0.011) | | NAB | 0.811 (0.005) | 0.785 (0.005) | |
| WBC | 0.607 (0.009) | 0.504 (0.011) | 0.490 (0.011) | WBC | 0.831 (0.004) | 0.816 (0.005) | 0.812 (0.005) |

Transition parameters:

| | $c$ | | | | $\eta$ | | |
|---|---|---|---|---|---|---|---|
| | ANZ | CBA | NAB | | ANZ | CBA | NAB |
| CBA | 0.550 (0.005) | | | CBA | 4.764 (0.280) | | |
| NAB | 0.567 (0.001) | 0.532 (0.008) | | NAB | 7.000 (−) | 4.514 (0.341) | |
| WBC | 0.555 (0.005) | 0.547 (0.004) | 0.549 (0.004) | WBC | 5.198 (0.308) | 4.761 (0.254) | 4.873 (0.294) |

Note: $\eta = \ln \gamma$; see Appendix A.2.

The steady increase of the correlations among the Big Four over time as well as the maintained high correlation state since the GFC could be linked to a few changes in the Australian financial sector. For instance, the time-varying correlation structure may reflect the four pillars policy, the financial concentration, which was further strengthened by the acquisition of the next largest banks by CBA and WBC in 2008. Further contributing factors that may have made the banks look more similar from the investors' point of view include the regulatory requirements brought by the implementation of Basel II, the easing of restrictions that directly impacted the home mortgage market that the Big Four now dominated and the stagnation of the housing credit market. Furthermore, since the GFC, all four banks have enjoyed substantial government support.

The fact that serious effort has been made to model the volatilities and correlations separately allows for observations on the timing and magnitude of those features without the cross-contamination occurring if covariances were examined instead. It is often noted that correlations increase during turbulent times. In the case of the Big Four, this is not exactly the case. The calm volatility period (from 2003 until late 2007) overlaps with the period of smoothly increasing correlations (16 months leading to early 2008). Furthermore, it is notable that the GFC has a tremendous impact on the volatilities, whereas the correlations have by then settled to their high levels and exhibit no further change.

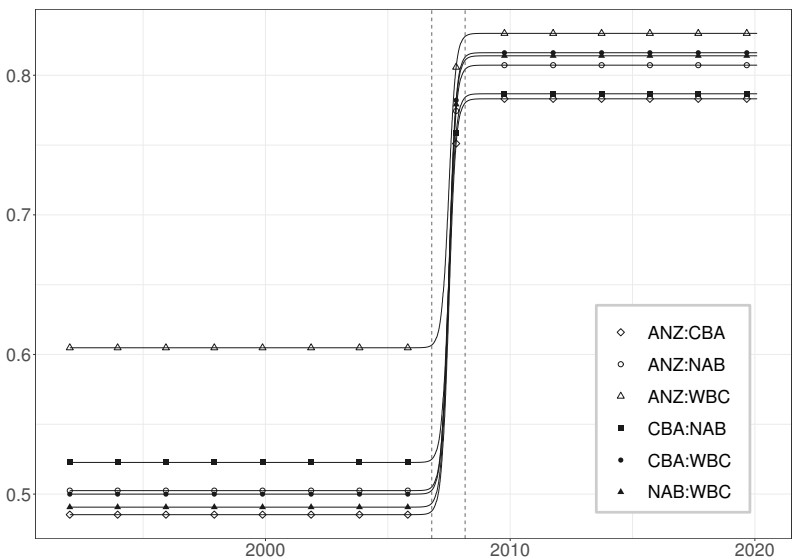

**Figure 8.** Estimated correlations. Vertical lines correspond to October 2007 and February 2008.

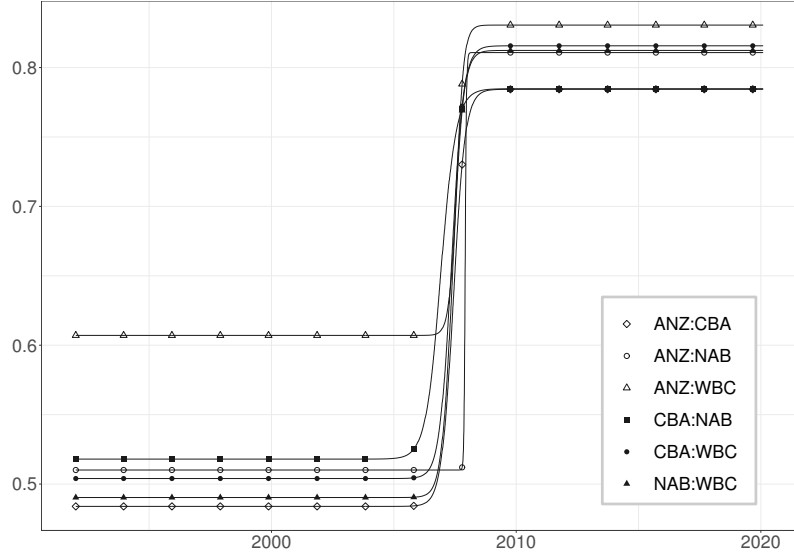

**Figure 9.** Estimated pairwise correlations.

## 8. Conclusions

In this paper, a data-driven modelling cycle for building MTV-GARCH models for asset returns was constructed and illustrated with an empirical example. The paper complements Silvennoinen and Teräsvirta (2021), which presented the asymptotic theory for this model, however, as already mentioned in the Introduction, does not contain any discussion on practical model-building issues. All three phases of the cycle: model specification, estimation, and evaluation, were covered. Specification includes testing the constancy of the GARCH equations against multiplicatively time-varying GARCH. This is a nonstandard testing problem as the MTV-GARCH model is not identified when the null hypothesis holds.

Furthermore, constructing the null model requires new techniques due to the fact that the conditional variance is not observed, and two such alternatives for solving this problem are presented. Simulations reported in an online appendix show that the proposed testing procedure had reasonable small-sample properties. In specifying the correlation structure,

the constancy of correlations has to be tested, and a relevant test for this nonstandard testing situation was developed.

The GARCH equations and the correlation structure, both nonlinear, were estimated jointly, and the technical details of this process were presented. Misspecification tests for the estimated model were derived to be used for the evaluation of the estimated model. The application to the four main Australian banks, the Big Four, demonstrated the use of the modelling cycle. The GARCH equations were found to be multiplicatively time-varying, and the correlations also changed over time. The estimation results indicated that the amplitude of the volatility clusters declines to a lower level around 2004 and temporarily rises during the GFC.

The (positive) correlations were found to be nonconstant, and they increased to a fairly high level already before the GFC. There is no single reason for this shift that is also established by estimating pairwise models for the six pairs of banks separately. Generally speaking, it may be argued that, whatever the aforementioned events before 2008 have meant to the ability of the banks to compete with each other, from the investors' viewpoint, they have become increasingly similar. Given their size, these four banks may represent a systemic risk to the Australian financial sector or the Australian economy in general.

Finally, the paper is accompanied by an R-package entitled `mtvgarch`, which is maintained in a private GitHub repository and contains all the econometric tools necessary for building MTV-GARCH models.

There are appendices contain additional material to the paper. Appendix A provides details of the TVV-model specification, the MTV-GARCH model evaluation, the test of constant correlations, and finally the test for an additional transition in the correlations. The simulations studies in Appendix B explore aspects of the specification and evaluation of the GARCH equations, and the size and sensitivity of the test of constant correlations. Appendix C presents the details of maximisation by parts. The estimated deterministic components of the Four Banks' transition equations are presented in Appendix D.

**Supplementary Materials:** The following supporting information can be downloaded at: https://www.mdpi.com/article/10.3390/econometrics11010005/s1.

**Author Contributions:** Conceptualization, A.D.H., A.S. and T.T.; methodology, T.T. and A.S.; software, A.S.; validation, A.S.; formal analysis, A.S. and T.T.; data curation, A.D.H.; writing A.S. and T.T.; visualization, A.S. All authors have read and agreed to the published version of the manuscript.

**Funding:** This research received no external funding.

**Data Availability Statement:** Supplementary material that includes data for the application, source code for the estimation and simulations, as well as the MTVGARCH package version used in the production of this paper can be found at https://econ.au.dk/research/researchcentres/creates/research/creates-research-papers/supplementary-downloads/rp-2021-13 (accessed on 26 January 2023).

**Acknowledgments:** This research was supported by the Center for Research in Econometric Analysis of Time Series (CREATES). An earlier version of this paper with the title 'Four Australian Banks and the Multivariate Time-Varying Smooth Transition Correlation GARCH model' appeared as CREATES Research Papers No. 2021–2013. Part of the work was conducted when the third author was visiting School of Economics and Finance of Queensland University of Technology, Brisbane, whose kind hospitality is gratefully acknowledged. We would also like to thank Glen Wade for their work with the R-package. Material from this paper was presented at the 26th Annual Symposium of the Society for Nonlinear Dynamics and Econometrics, Tokyo, March 2018; the workshop 'Frontiers in Econometrics', Queensland University of Technology, Brisbane, July 2018; the Quantitative Methods in Finance 2018 Conference (UTS), Sydney, December 2018; the 13th International Conference of the ERCIM WG on Computational and Methodological Statistics, London, December 2020; the International Association of Applied Econometrics Conference, Rotterdam, July 2021; and the Workshop on Financial Econometrics, Örebro University, November 2021. Comments from the participants are gratefully acknowledged. The responsibility for any errors and shortcomings in this work remains ours.

**Conflicts of Interest:** The authors declare no conflict of interest.

## Appendix A. Test Statistics

*Appendix A.1. Test Statistic for TVV-Model Specification*

In order to specify $g_t$, we not only test constancy but even specify the number of transitions before estimating the GARCH component of the model. Amado and Teräsvirta (2013) showed that maximum likelihood estimators of the corresponding time-varying variance (TVV) model, assuming that there is no conditional heteroskedasticity, are consistent and asymptotically normal. This forms the base for constructing Lagrange multiplier type tests for testing $r$ against $r + 1$ transitions. For notational simplicity, consider testing one transition against two. Omitting the subscript $i$ for simplicity, the TVV model is (9) with $h_t = 1$, and

$$g_t = \delta_0 + \delta_1 G_1(t/T, \gamma_1, c_1) + \delta_2 G_2(t/T, \gamma_2, c_2), \ \gamma_i > 0, \ i = 1, 2.$$

The null hypothesis is $\gamma_2 = 0$—in which case, $G_2(t/T, \gamma_2, c_2) \equiv 1/2$. To circumvent the identification problem (the model with one transition is only identified when the alternative $\gamma_2 > 0$ is true), we follow Luukkonen et al. (1988) and approximate the second transition by a third-order Taylor expansion around the null hypothesis. After reparametrisation, this yields

$$g_t = \delta_0^{*0} + \delta_1 G_1(t/T, \gamma_1, c_1) + \psi_1 t/T + \psi_2 (t/T)^2 + \psi_3 (t/T)^3, \ \gamma_1 > 0. \quad \text{(A1)}$$

We may call (9) with (A1) the auxiliary TVV model. The parameters $\psi_i = \gamma_2 \widetilde{\psi}_i$, where $\widetilde{\psi}_i \neq 0$, $i = 1, 2, 3$. The new null hypothesis in (A1) equals $\text{H}_0'$: $\psi_1 = \psi_2 = \psi_3 = 0$. The remainder term of the expansion can be ignored because when we construct a Lagrange multiplier test, the model is only estimated under $\text{H}_0$ (or $\text{H}_0'$), and, under this hypothesis, the order of the Taylor expansion equals zero. The remainder is present only under the alternative, and thus ignoring it when $\text{H}_0$ is valid does not affect the asymptotic size of the test. It does make a positive contribution to the power of the test when $\text{H}_0$ does not hold.

Assume (again, for notational simplicity) that $K_1 = 1$ in (A1), so $c_1 = c_1$ (a scalar). The log-likelihood for observation $t$ of the auxiliary TVV model equals

$$\ell_t = k - (1/2) \ln g_t - (1/2) \frac{\varepsilon_t^2}{g_t}$$

and the corresponding element of the score is

$$\frac{\partial \ell_t}{\partial \boldsymbol{\theta}_1} = \frac{1}{2} \left( \frac{\varepsilon_t^2}{g_t} - 1 \right) \frac{1}{g_t} \frac{\partial g_t}{\partial \boldsymbol{\theta}_1}, \quad \text{(A2)}$$

where $\boldsymbol{\theta}_1 = (\delta_0^{*0}, \delta_1, \gamma_1, c_1, \psi_1, \psi_2, \psi_3)'$. Denoting $G_1(t/T) = G_1(t/T, \gamma_1, c_1)$, the partial derivative in (A2) is $\partial g_t / \partial \boldsymbol{\theta}_1 = (g_1'(t/T), \boldsymbol{\tau}_t')'$ where

$$g_1(t/T) = (1, G_1(t/T), G_{1\gamma}(t/T), G_{1c}(t/T) G_{1\gamma}(t/T))'$$

with $G_{1\gamma}(t/T) = G_1(t/T)(1 - G_1(t/T))(t/T - c_1)$, $G_{1c}(t/T) = -\gamma_1 G_1(t/T)(1 - G_1(t/T))$, and $\boldsymbol{\tau}_t = (t/T, (t/T)^2, (t/T)^3)$. Define the true parameter vector under $\text{H}_0$ as $\boldsymbol{\theta}_1^0 = (\delta_0^{*0}, \delta_1^0, \gamma_1^0, c_1^0, 0, 0, 0)'$. If $z_t$ is normally distributed, the corresponding element of the information matrix under $\text{H}_0$ has the form

$$\boldsymbol{B}_t = \frac{1}{4} \mathsf{E} \left( \frac{\varepsilon_t^2}{g_t} - 1 \right)^2 \begin{bmatrix} \boldsymbol{B}_{11t} & \boldsymbol{B}_{12t} \\ \boldsymbol{B}_{21t} & \boldsymbol{B}_{22t} \end{bmatrix} = \frac{1}{2} \begin{bmatrix} \boldsymbol{B}_{11t} & \boldsymbol{B}_{12t} \\ \boldsymbol{B}_{21t} & \boldsymbol{B}_{22t} \end{bmatrix},$$

where, letting $g_t^0 = g^0(t/T) = \delta_0^{*0} + \delta_1^0 G_1(t/T, \gamma_1^0, c_1^0)$ and denoting $G_1^0(t/T) = G_1(t/T, \gamma_1^0, c_1^0)$,

$$\boldsymbol{B}_{11t} = \frac{1}{2(g^0(t/T))^2}\boldsymbol{g}_1^0(t/T)\boldsymbol{g}_1^0(t/T)', \quad \boldsymbol{B}_{12t} = \frac{1}{2(g^0(t/T))^2}\boldsymbol{g}_1^0(t/T)\boldsymbol{\tau}_t'$$

and

$$\boldsymbol{B}_{22t} = \frac{1}{2(g^0(t/T))^2}\boldsymbol{\tau}_t\boldsymbol{\tau}_t'.$$

Let

$$\boldsymbol{g}_1^0(r) = (1, G_1^0(r), G_{1\gamma}^0(r), G_{1c}^0(r))'$$

and $\boldsymbol{r} = (r, r^2, r^3)'$. We state the following lemma:

**Lemma A1.** *Under the null hypothesis and assuming $z_t \sim iid\mathcal{N}(0,1)$, the information matrix*

$$\boldsymbol{B} = \lim_{T\to\infty} \frac{1}{2T}\sum_{t=1}^{T}\begin{bmatrix} \boldsymbol{B}_{11t} & \boldsymbol{B}_{12t} \\ \boldsymbol{B}_{21t} & \boldsymbol{B}_{22t} \end{bmatrix} = \frac{1}{2}\begin{bmatrix} \boldsymbol{B}_{11} & \boldsymbol{B}_{12} \\ \boldsymbol{B}_{21} & \boldsymbol{B}_{22} \end{bmatrix},$$

*where*

$$\boldsymbol{B}_{11} = \frac{1}{2}\int_0^1 (g^0(r))^{-2}\boldsymbol{g}_1^0(r)\boldsymbol{g}_1^0(r)'dr, \quad \boldsymbol{B}_{12} = \frac{1}{2}\int_0^1 (g^0(r))^{-2}\boldsymbol{g}_1^0(r)\boldsymbol{r}'dr$$

*and*

$$\boldsymbol{B}_{22} = \frac{1}{2}\int_0^1 (g^0(r))^{-2}\boldsymbol{r}\boldsymbol{r}'dr.$$

**Proof.** The 'sample' information matrix with $T$ observations equals

$$\frac{1}{4T}\sum_{t=1}^{T}\boldsymbol{B}_t = \frac{1}{4T}\sum_{t=1}^{T}\begin{bmatrix} \boldsymbol{B}_{11t} & \boldsymbol{B}_{12t} \\ \boldsymbol{B}_{21t} & \boldsymbol{B}_{22t} \end{bmatrix}.$$

Consider the $(1,2)$ element of $\boldsymbol{B}_{11(T)} = (1/T)\sum_{t=1}^{T}\boldsymbol{B}_{11t}$:

$$[\boldsymbol{B}_{11(T)}]_{12} = (1/T)\sum_{t=1}^{T}(g^0(t/T))^{-2}G_1^0(t/T),$$

which is an average of $T$ values of the logistic cumulative distribution function. Let $[Tr] = t$ be the integer closest to $t$. Then,

$$\begin{aligned}
(1/T)\sum_{t=1}^{T}(g^0(t/T))^{-2}G_1^0(t/T) &= \sum_{t=1}^{T}\int_{t/T}^{(t+1)/T}(g^{*0}([Tr]/T))^{-2}G_1^0([Tr]/T)\mathrm{d}r \\
&= \int_{1/T}^{(T+1)/T}(g^{*0}([Tr]/T))^{-2}G_1^0([Tr]/T)\mathrm{d}r \\
&\to \int_0^1 (g^{*0}(r))^{-2}G_1^0(r)\mathrm{d}r
\end{aligned}$$

as $T \to \infty$. The other elements of $\boldsymbol{B}_{11} = \lim_{T\to\infty}\boldsymbol{B}_{11(T)}$, are derived in a similar fashion. In matrix form,

$$\boldsymbol{B}_{11} = \frac{1}{2}\int_0^1 (g^{*0}(r))^{-2}\boldsymbol{g}_1^0(r)\boldsymbol{g}_1^0(r)'\mathrm{d}r.$$

The blocks $\boldsymbol{B}_{12}$ and $\boldsymbol{B}_{22}$ are obtained similarly. □

Since the maximum likelihood estimators of the parameters of the auxiliary TVV model under $H_0$ are consistent, we may construct the LM test for the hypothesis $H_0'$: $\boldsymbol{\psi} = (\psi_1, \psi_2, \psi_3)' = \mathbf{0}$. Denoting the relevant block of the score by

$$s_2(\widehat{\boldsymbol{\theta}}_1) = \frac{1}{2T} \sum_{t=1}^{T} (\frac{\varepsilon_t^2}{\widehat{g}_t} - 1) \frac{1}{\widehat{g}_t} \frac{\partial g_t}{\partial \boldsymbol{\psi}},$$

where $\partial g_t / \partial \boldsymbol{\psi} = \boldsymbol{\tau}_t$ and

$$\widehat{g}_t = \widehat{\delta}_0 + \widehat{\delta}_1 (1 + \exp\{-\widehat{\gamma}_1 (t/T - \widehat{c}_1)\})^{-1}.$$

Then, assuming $z_t = \varepsilon_t / g_t^{1/2}$ is standard normal under $H_0$, the test statistic has the following form:

$$LM_T^1 = (T/2) s_2'(\widehat{\boldsymbol{\theta}}_1)(\boldsymbol{B}_{22} - \boldsymbol{B}_{21} \boldsymbol{B}_{11}^{-1} \boldsymbol{B}_{12})^{-1} s_2(\widehat{\boldsymbol{\theta}}_1), \tag{A3}$$

where $\widehat{\boldsymbol{\theta}}_1 = (\widehat{\delta}_0^*, \widehat{\delta}_1, \widehat{\gamma}_1, \widehat{c}_1, 0, 0, 0)'$; see, for example, Godfrey (1988, p. 14). In order to make (A3) operational, the blocks of $\boldsymbol{B}$ are replaced by their consistent counterparts.

When constancy of the error variance is tested against a single transition, $g_t \equiv \delta_0$, $g_1(t/T) = 1$ (scalar), and $\partial g_t / \partial \boldsymbol{\psi} = \boldsymbol{\tau}_t$ as before. Then, $\boldsymbol{B}_{11} = (2(\delta_0^0)^2)^{-1}$,

$$\boldsymbol{B}_{12} = \frac{1}{2(\delta_0^0)^2} \int_0^1 \boldsymbol{r}' \mathrm{d}r \text{ and } \boldsymbol{B}_{22} = \frac{1}{2(\delta_0^0)^2} \int_0^1 \boldsymbol{r} \boldsymbol{r}' \mathrm{d}r.$$

The test statistic (A3) becomes

$$LM_T^0 = \frac{T(\delta_0^0)^2}{2\widehat{\delta}_0^2} s_2'(\widehat{\boldsymbol{\theta}}_1)(\boldsymbol{B}_{22} - \boldsymbol{B}_{21} \boldsymbol{B}_{11}^{-1} \boldsymbol{B}_{12})^{-1} s_2(\widehat{\boldsymbol{\theta}}_1), \tag{A4}$$

where

$$s_2(\widehat{\boldsymbol{\theta}}_1) = \frac{1}{2T} \sum_{t=1}^{T} (\frac{\varepsilon_t^2}{\widehat{\delta}_0} - 1) \boldsymbol{\tau}_t.$$

When the elements of the covariance matrix are replaced by their consistent estimators in (A4), the ratio $(\delta_0^0)^2 / \widehat{\delta}_0^2$ equals unity.

As already mentioned, conditional heteroskedasticity is ignored in setting up the test. For this reason, the test statistic (A3) is likely to be size distorted when applied to financial time series of sufficiently high frequency—that is, when GARCH-type volatility clustering is present. In applications, its size has to be adjusted by calibrating its distribution to reflect the persistence of the GARCH effect present in the data. This is the topic of discussion in Appendix B.1.

*Appendix A.2. Test Statistic for MTV-GARCH Model Evaluation*

In this section, the evaluation tests of the univariate MTV-GARCH equations are presented in an easy to implement fashion. The full details can be found in Amado and Teräsvirta (2017).

The test statistic is computed based on the following components: $\widehat{\zeta}_t$, $\boldsymbol{r}_{1t}$, and $\boldsymbol{r}_{2t}$. $\widehat{\zeta}_t = \varepsilon_t / \sqrt{\widehat{h}_t \widehat{g}_t}$ are the residuals, $\boldsymbol{r}_{1t}$ contains the derivatives of the functions $h_t$ and $g_t$ with respect to the parameters that govern the MTV-GARCH model under the null, $\boldsymbol{\theta}_g$ and $\boldsymbol{\theta}_h$:

$$\boldsymbol{r}_{1t} = (\widehat{g}_t^{-1} \frac{\partial g_t}{\partial \widehat{\boldsymbol{\theta}}_g} + \widehat{h}_t^{-1} \frac{\partial h_t}{\partial \widehat{\boldsymbol{\theta}}_g}, \widehat{h}_t^{-1} \frac{\partial h_t}{\partial \widehat{\boldsymbol{\theta}}_h}),$$

evaluated at the estimated parameters $\widehat{\boldsymbol{\theta}}_g$ and $\widehat{\boldsymbol{\theta}}_h$. These are recursively calculated, and depend on the prevailing MTV-GARCH model under the null. For example, one could have $g_t = \delta_0 + \delta_1 G_1(t/T, \gamma_1, c_1) + \delta_2 G_2(t/T, \gamma_2, (c_{21}, c_{22})')$ and $h_t = \alpha_0 + \alpha_1 \varepsilon_{t-1}^2 / g_{t-1} +$

$\kappa_1 I(\varepsilon_{t-1} < 0)\varepsilon_{t-1}^2/g_{t-1} + \beta_1 h_{t-1}$. Here, $\boldsymbol{\theta}_g = (\delta_1, \gamma_1, c_1, \delta_2, \gamma_2, c_{21}, c_{22})'$ and $\boldsymbol{\theta}_h = (\alpha_0, \alpha_1, \kappa_1, \beta_1)'$. Then,

$$\frac{\partial g_t}{\partial \boldsymbol{\theta}_g} = (G_1, \delta_1 \frac{\partial G_1}{\partial \gamma_1}, \delta_1 \frac{\partial G_1}{\partial c_1}, G_2, \delta_2 \frac{\partial G_2}{\partial \gamma_2}, \delta_2 \frac{\partial G_2}{\partial c_{21}}, \delta_2 \frac{\partial G_2}{\partial c_{22}})'$$

where

$$\frac{\partial G_1}{\partial \gamma_1} = G_1(1 - G_1)(t/T - c_1)$$

$$\frac{\partial G_1}{\partial c_1} = -G_1(1 - G_1)\gamma_1$$

$$\frac{\partial G_2}{\partial \gamma_2} = G_2(1 - G_2)(t/T - c_{21})(t/T - c_{22})$$

$$\frac{\partial G_2}{\partial c_{21}} = -G_2(1 - G_2)\gamma_2(t/T - c_{22})$$

$$\frac{\partial G_2}{\partial c_{22}} = -G_2(1 - G_2)\gamma_2(t/T - c_{21}).$$

The GARCH equation derivatives are formed recursively as

$$\frac{\partial h_t}{\partial \boldsymbol{\theta}_g} = -g_t^{-1}(\alpha_1 \varepsilon_{t-1}^2/g_{t-1} + \kappa_1 I(\varepsilon_{t-1} < 0)\varepsilon_{t-1}^2/g_{t-1})\frac{\partial g_{t-1}}{\partial \boldsymbol{\theta}_g} + \beta_1 \frac{\partial h_{t-1}}{\partial \boldsymbol{\theta}_g}$$

and

$$\frac{\partial h_t}{\partial \boldsymbol{\theta}_h} = (\varepsilon_{t-1}^2/g_{t-1}, I(\varepsilon_{t-1} < 0)\varepsilon_{t-1}^2/g_{t-1}, h_{t-1})' + \beta_1 \frac{\partial h_{t-1}}{\partial \boldsymbol{\theta}_h}$$

From this example, it should be easy to extend the null model to include more additive deterministic terms and/or have a higher order GARCH equation with or without asymmetric terms.

One extension regarding the deterministic part should be mentioned. It is often convenient to replace the slope parameter $\gamma$ with $e^\eta$. In this case, $\boldsymbol{\theta}_g = (\delta_1, \eta_1, c_1, \delta_2, \eta_2, c_{21}, c_{22})'$, and

$$\frac{\partial G_1}{\partial \eta_1} = G_1(1 - G_1)e^{\eta_1}(t/T - c_1)$$

$$\frac{\partial G_1}{\partial c_1} = -G_1(1 - G_1)e^{\eta_1}$$

$$\frac{\partial G_2}{\partial \eta_2} = G_2(1 - G_2)e^{\eta_2}(t/T - c_{21})(t/T - c_{22})$$

$$\frac{\partial G_2}{\partial c_{21}} = -G_2(1 - G_2)e^{\eta_2}(t/T - c_{22})$$

$$\frac{\partial G_2}{\partial c_{22}} = -G_2(1 - G_2)e^{\eta_2}(t/T - c_{21}).$$

Vector $\boldsymbol{r}_{2t}$ contains the derivatives of the misspecified part. Details in the most commonly encountered situations will be given shortly. The number of variables (columns) in $\boldsymbol{r}_{2t}$ defines the degrees of freedom in the $\chi^2$-distribution for the test statistic under the null.

Given the three components, the LM-test is performed as follows:

1. Compute the $SSR_0 = \sum_{t=1}^T (\hat{\zeta}_t^2 - 1)^2$.
2. Regress $\hat{\zeta}_t^2 - 1$ on $(\boldsymbol{r}_{1t}, \boldsymbol{r}_{2t})$, and form the sum of squared residuals $SSR_1$.
3. Compute the test statistic $LM = T\frac{SSR_0 - SSR_1}{SSR_0}$.

The robust version that does not rely on the normality of the error term is formed as follows:

1. Regress $r_{2t}$ on $r_{1t}$ and obtain residuals $w_t$. When $r_{2t}$ has more than one variable, run the regression for each of them separately and, thereby, obtain a set of residuals $w_t$.
2. Regress $\mathbf{1}$ on $(\hat{\zeta}_t^2 - 1)w_t$ and form the sum of squared residuals $SSR$.
3. Compute the test statistic $LM_R = T - SSR$.

The first case seeks to find evidence of misspecification of the determininstic part of the MTV-GARCH model. The conditional variance is of the form

$$\sigma_t^2 = h_t(g_t + f_t),$$

where the additive term $f_t$ is zero under the null of the model being correctly specified. The case that we consider here is the one of testing $r$ against $r + 1$ transitions in the deterministic part. The additive term is linearised and reparameterised, after which, it becomes

$$f_t = \delta_0^* + \delta_1^* t/T + \delta_2^* (t/T)^2 + \delta_3^* (t/T)^3.$$

The derivative component for the alternative is then

$$r_{2t} = \hat{g}_t^{-1}(1, t/T, (t/T)^2, (t/T)^3).$$

The second case addresses misspecification in the GARCH part:

$$\sigma_t^2 = (h_t + f_t)g_t,$$

where the additive term $f_t$ is again zero under the null. A common scenario is when $f_t$ may increase either the ARCH or the GARCH order (but not both). An example of the former is GARCH(1,1) vs. GARCH(2,1), in which case, $f_t = \alpha_2 \varepsilon_{t-2}^2/g_{t-2}$, and therefore

$$r_{2t} = \hat{h}_t^{-1} \varepsilon_{t-2}^2/\hat{g}_{t-2}.$$

If the model is a GJR one, and the potential increase in the order of the ARCH term extends to the asymmetric terms as well, then $f_t = \alpha_2 \varepsilon_{t-2}^2/g_{t-2} + \kappa_2 I(\varepsilon_{t-2} < 0)\varepsilon_{t-2}^2/g_{t-2}$, and

$$r_{2t} = \hat{h}_t^{-1}(\varepsilon_{t-2}^2/\hat{g}_{t-2}, I(\varepsilon_{t-2} < 0)\varepsilon_{t-2}^2/\hat{g}_{t-2}).$$

An example of the latter is GARCH(1,1) vs. GARCH(2,1), which leads to $f_t = \beta_2 h_{t-2}$, and thus

$$r_{2t} = \hat{h}_t^{-1}\hat{h}_{t-2}.$$

The third case is the test of no remaining ARCH. This is a test against multiplicative misspecification,

$$\sigma_t^2 = h_t g_t f_t,$$

where $f_t = 1$ under the null. If the alternative is that there is ARCH of order $m$ left unaccounted for, then

$$r_{2t} = (\hat{\zeta}_{t-1}^2, \dots, \hat{\zeta}_{t-m}^2).$$

*Appendix A.3. Test of Constant Correlations*

The log-likelihood of the auxiliary MTV model for observation $t$ assuming $K = 2$ equals

$$\ln f_A(\boldsymbol{\zeta}_t|\boldsymbol{\theta}) = -(1/2)\sum_{i=1}^N \ln g_{it} - (1/2)\sum_{i=1}^N \ln h_{it} - (1/2)\ln|\boldsymbol{P}_{At}|$$
$$- (1/2)\varepsilon_t'\{\boldsymbol{S}_t \boldsymbol{D}_t \boldsymbol{P}_{At} \boldsymbol{D}_t \boldsymbol{S}_t\}^{-1}\varepsilon_t,$$

where

$$\boldsymbol{P}_{At} = \boldsymbol{P}_{(A0)} + (t/T)\boldsymbol{P}_{(A1)} + (t/T)^2\boldsymbol{P}_{(A2)}$$

and $g_{it} = \delta_{i0} + \delta_{i1}G_{i1}(t/T, \gamma_{i1}, c_{i1})$; only one transition for notational simplicity, and $h_{it}$ is as in (4). The first sub-block of the score corresponding to the deterministic variance component under $H_0$ becomes

$$s_t(\boldsymbol{\theta}_{gi}) = -\frac{1}{2}(g_{it}^{-1}\frac{\partial g_{it}}{\partial \boldsymbol{\theta}_{gi}} + h_{it}^{-1}\frac{\partial h_{it}}{\partial \boldsymbol{\theta}_{gi}})(1 - \boldsymbol{e}_i' \boldsymbol{P}_{(A0)}^{-1} \boldsymbol{z}_t \boldsymbol{z}_t' \boldsymbol{e}_i),$$

where $\boldsymbol{e}_i = (\boldsymbol{0}_{i-1}', 1, \boldsymbol{0}_{N-i}')'$, $i = 1, \ldots, N$, and $\boldsymbol{0}_0$ is an empty set. The sub-block corresponding to the GARCH parameters under $H_0$ is

$$s_t(\boldsymbol{\theta}_{hi}) = -\frac{1}{2}(h_{it}^{-1}\frac{\partial h_{it}}{\partial \boldsymbol{\theta}_{hi}})(1 - \boldsymbol{e}_i' \boldsymbol{P}_{(A0)}^{-1} \boldsymbol{z}_t \boldsymbol{z}_t' \boldsymbol{e}_i),$$

$i = 1, \ldots, N$. The remaining sub-blocks under $H_0$ equal

$$\begin{aligned} s_t(\boldsymbol{\rho}_{Aj}) &= -\frac{1}{2}\frac{\partial \text{vec}(\boldsymbol{P}_{At})'}{\partial \boldsymbol{\rho}_{Aj}}\{\text{vec}(\boldsymbol{P}_{(A0)}^{-1}) - (\boldsymbol{P}_{(A0)}^{-1} \otimes \boldsymbol{P}_{(A0)}^{-1})\text{vec}(\boldsymbol{z}_t \boldsymbol{z}_t')\} \\ &= -\frac{1}{2}(t/T)^j \boldsymbol{U}'\{\text{vec}(\boldsymbol{P}_{(A0)}^{-1}) - (\boldsymbol{P}_{(A0)}^{-1} \otimes \boldsymbol{P}_{(A0)}^{-1})\text{vec}(\boldsymbol{z}_t \boldsymbol{z}_t')\}, \end{aligned}$$

$j = 0, 1, 2$, where $\boldsymbol{U} = \partial \text{vec}(\boldsymbol{P}_{(Aj)})/\partial \boldsymbol{\rho}_{Aj}'$ consists of zeroes and ones and is identical for all $j$. The $N^2 \times N(N-1)/2$ matrix $\boldsymbol{U}$ is a column-wise collection of vectorised indicator matrices that identify the locations of the particular correlation parameters within the matrix $\boldsymbol{P}_{At}$. For example, the first correlation parameter in $\boldsymbol{\rho}_{Aj}$ is located in positions (2,1) and (1,2) in $\boldsymbol{P}_{At}$. An indicator matrix corresponding to this parameter has ones in those positions and zeros elsewhere. This vectorised indicator matrix is then the first column of matrix $\boldsymbol{U}$ and so on. Consequently, the $3N(N-1)/2 \times N^2$ matrix $\partial \text{vec}(\boldsymbol{P}_{At})'/\partial \boldsymbol{\rho}_A$ equals

$$\frac{\partial \text{vec}(\boldsymbol{P}_{At})'}{\partial \boldsymbol{\rho}_A} = \begin{bmatrix} 1 \\ (t/T) \\ (t/T)^2 \end{bmatrix} \otimes \boldsymbol{U}'.$$

The information matrix for observation $t$ under $H_0$ is quite similar to, but simpler than, the corresponding one in Silvennoinen and Teräsvirta (2021). In order to give the matrix a proper expression, we need the commutation matrix $\boldsymbol{K}$, an $N^2 \times N^2$ matrix whose $(i, j)$ block equals $\boldsymbol{e}_j \boldsymbol{e}_i'$—that is, $[\boldsymbol{K}]_{ij} = \boldsymbol{e}_j \boldsymbol{e}_i'$; see, for example, Lütkepohl (1996, pp. 115–18). Let the superscript 0 indicate that the corresponding entity is evaluated under $H_0$ (for example, $g_{it}^0$ equals $g_{it}|_{H_0}$, and $\partial g_{it}^0/\partial \boldsymbol{\theta}_{gi}$ equals $\partial g_{it}/\partial \boldsymbol{\theta}_{gi}|_{H_0}$). The matrix is defined in the following lemma.

**Lemma A2.** *The expectations of the nine blocks of the information matrix at (rescaled) time $t/T$ under $H_0$: $\boldsymbol{\rho}_{A1} = \boldsymbol{\rho}_{A2} = \boldsymbol{0}_{N(N-1)/2}$ are*

$$\boldsymbol{B}_t^0 = \mathsf{E}s_t(\boldsymbol{\theta}^0)s_t'(\boldsymbol{\theta}^0) = \mathsf{E}\begin{bmatrix} s_t(\boldsymbol{\theta}_g^0)s_t'(\boldsymbol{\theta}_g^0) & s_t(\boldsymbol{\theta}_g^0)s_t'(\boldsymbol{\theta}_h^0) & s_t(\boldsymbol{\theta}_g^0)s_t'(\boldsymbol{\rho}_A) \\ s_t(\boldsymbol{\theta}_h^0)s_t'(\boldsymbol{\theta}_g^0) & s_t(\boldsymbol{\theta}_h^0)s_t'(\boldsymbol{\theta}_h^0) & s_t(\boldsymbol{\theta}_h^0)s_t'(\boldsymbol{\rho}_A) \\ s_t(\boldsymbol{\rho}_A)s_t'(\boldsymbol{\theta}_g^0) & s_t(\boldsymbol{\rho}_A)s_t'(\boldsymbol{\theta}_h^0) & s_t(\boldsymbol{\rho}_A)s_t'(\boldsymbol{\rho}_A) \end{bmatrix}.$$

*The $(i, j)$ sub-block of $\boldsymbol{B}_{11t} = \mathsf{E}s_t(\boldsymbol{\theta}_g^0)s_t'(\boldsymbol{\theta}_g^0)$, $i \neq j$, equals*

$$\begin{aligned} [\boldsymbol{B}_{11t}]_{ij} &= \mathsf{E}s_t(\boldsymbol{\theta}_{gi}^0)s_t'(\boldsymbol{\theta}_{gj}^0) \\ &= \frac{1}{4}(\frac{1}{g_{it}^0}\frac{\partial g_{it}^0}{\partial \boldsymbol{\theta}_{gi}} + \frac{1}{h_{it}^0}\frac{\partial h_{it}^0}{\partial \boldsymbol{\theta}_{gi}})(\frac{1}{g_{jt}^0}\frac{\partial g_{jt}^0}{\partial \boldsymbol{\theta}_{gj}'} + \frac{1}{h_{jt}^0}\frac{\partial h_{jt}^0}{\partial \boldsymbol{\theta}_{gj}'})\boldsymbol{e}_i' \boldsymbol{P}_{(A0)}^{-1} \boldsymbol{e}_j \boldsymbol{e}_i' \boldsymbol{P}_{(A0)} \boldsymbol{e}_j. \end{aligned}$$

When $i = j$,

$$[\boldsymbol{B}_{11t}]_{ii} = \mathsf{E}\boldsymbol{s}_t(\boldsymbol{\theta}_{gi}^0)\boldsymbol{s}_t'(\boldsymbol{\theta}_{gi}^0)$$
$$= \frac{1}{4}\left(\frac{1}{g_{it}^0}\frac{\partial g_{it}^0}{\partial \boldsymbol{\theta}_{gi}} + \frac{1}{h_{it}^0}\frac{\partial h_{it}^0}{\partial \boldsymbol{\theta}_{gi}}\right)\left(\frac{1}{g_{it}^0}\frac{\partial g_{it}^0}{\partial \boldsymbol{\theta}_{gi}'} + \frac{1}{h_{it}^0}\frac{\partial h_{it}^0}{\partial \boldsymbol{\theta}_{gi}'}\right)(1 + \boldsymbol{e}_i'\boldsymbol{P}_{(A0)}^{-1}\boldsymbol{e}_i).$$

The $(i, j)$ sub-block of $\boldsymbol{B}_{22t} = \mathsf{E}\boldsymbol{s}_t(\boldsymbol{\theta}_h^0)\boldsymbol{s}_t'(\boldsymbol{\theta}_h^0)$, $i \neq j$, equals

$$[\boldsymbol{B}_{22t}]_{ij} = \mathsf{E}\boldsymbol{s}_t(\boldsymbol{\theta}_{hi}^0)\boldsymbol{s}_t'(\boldsymbol{\theta}_{hj}^0)$$
$$= \frac{1}{4}\left(\frac{1}{h_{it}^0}\frac{\partial h_{it}^0}{\partial \boldsymbol{\theta}_{hi}}\right)\left(\frac{1}{h_{jt}^0}\frac{\partial h_{jt}^0}{\partial \boldsymbol{\theta}_{hj}'}\right)\boldsymbol{e}_i'\boldsymbol{P}_{(A0)}^{-1}\boldsymbol{e}_j\boldsymbol{e}_i'\boldsymbol{P}_{(A0)}\boldsymbol{e}_j.$$

When $i = j$,

$$[\boldsymbol{B}_{22t}]_{ii} = \mathsf{E}\boldsymbol{s}_t(\boldsymbol{\theta}_{hi}^0)\boldsymbol{s}_t'(\boldsymbol{\theta}_{hi}^0)$$
$$= \frac{1}{4}\left(\frac{1}{h_{it}^0}\frac{\partial h_{it}^0}{\partial \boldsymbol{\theta}_{hi}}\right)\left(\frac{1}{h_{it}^0}\frac{\partial h_{it}^0}{\partial \boldsymbol{\theta}_{hi}'}\right)(1 + \boldsymbol{e}_i'\boldsymbol{P}_{(A0)}^{-1}\boldsymbol{e}_i).$$

The $(i, j)$ sub-block of $\boldsymbol{B}_{12t} = \mathsf{E}\boldsymbol{s}_t(\boldsymbol{\theta}_g^0)\boldsymbol{s}_t'(\boldsymbol{\theta}_h^0)$, $i \neq j$, equals

$$[\boldsymbol{B}_{12t}]_{ij} = \mathsf{E}\boldsymbol{s}_t(\boldsymbol{\theta}_{gi}^0)\boldsymbol{s}_t'(\boldsymbol{\theta}_{hj}^0)$$
$$= \frac{1}{4}\left(\frac{1}{g_{it}^0}\frac{\partial g_{it}^0}{\partial \boldsymbol{\theta}_{gi}} + \frac{1}{h_{it}^0}\frac{\partial h_{it}^0}{\partial \boldsymbol{\theta}_{gi}}\right)\left(\frac{1}{h_{jt}^0}\frac{\partial h_{jt}^0}{\partial \boldsymbol{\theta}_{hj}'}\right)\boldsymbol{e}_i'\boldsymbol{P}_{(A0)}^{-1}\boldsymbol{e}_j\boldsymbol{e}_i'\boldsymbol{P}_{(A0)}\boldsymbol{e}_j.$$

When $i = j$,

$$[\boldsymbol{B}_{12t}]_{ii} = \mathsf{E}\boldsymbol{s}_t(\boldsymbol{\theta}_{gi}^0)\boldsymbol{s}_t'(\boldsymbol{\theta}_{hi}^0)$$
$$= \frac{1}{4}\left(\frac{1}{g_{it}^0}\frac{\partial g_{it}^0}{\partial \boldsymbol{\theta}_{gi}} + \frac{1}{h_{it}^0}\frac{\partial h_{it}^0}{\partial \boldsymbol{\theta}_{gi}}\right)\left(\frac{1}{h_{it}^0}\frac{\partial h_{it}^0}{\partial \boldsymbol{\theta}_{hi}'}\right)(1 + \boldsymbol{e}_i'\boldsymbol{P}_{(A0)}^{-1}\boldsymbol{e}_i).$$

Furthermore, the $(i, j)$ sub-block of $\mathsf{E}\boldsymbol{s}_t(\boldsymbol{\theta}_g^0)\boldsymbol{s}_t'(\boldsymbol{\rho}_A)$ equals

$$[\boldsymbol{B}_{13t}]_{ij} = \mathsf{E}\boldsymbol{s}_t(\boldsymbol{\theta}_{gi}^0)\boldsymbol{s}_t'(\boldsymbol{\rho}_{Aj})$$
$$= \frac{1}{4}(t/T)^j\left(\frac{1}{g_{it}^0}\frac{\partial g_{it}^0}{\partial \boldsymbol{\theta}_{gi}} + \frac{1}{h_{it}^0}\frac{\partial h_{it}^0}{\partial \boldsymbol{\theta}_{gi}}\right)\{(\boldsymbol{e}_i \otimes \boldsymbol{e}_i)'(\boldsymbol{P}_{(A0)}^{-1} \otimes \boldsymbol{I}_N) + (\boldsymbol{e}_i \otimes \boldsymbol{e}_i)'(\boldsymbol{I}_N \otimes \boldsymbol{P}_{(A0)}^{-1})\}\boldsymbol{U},$$

$i = 1, \ldots, N$; $j = 0, 1, 2$. The $(i, j)$ sub-block of $\mathsf{E}\boldsymbol{s}_t(\boldsymbol{\theta}_h^0)\boldsymbol{s}_t'(\boldsymbol{\rho}_A)$ equals

$$[\boldsymbol{B}_{23t}]_{ij} = \mathsf{E}\boldsymbol{s}_t(\boldsymbol{\theta}_{hi}^0)\boldsymbol{s}_t'(\boldsymbol{\rho}_{Aj})$$
$$= \frac{1}{4}(t/T)^j\left(\frac{1}{h_{it}^0}\frac{\partial h_{it}^0}{\partial \boldsymbol{\theta}_{hi}}\right)\{(\boldsymbol{e}_i \otimes \boldsymbol{e}_i)'(\boldsymbol{P}_{(A0)}^{-1} \otimes \boldsymbol{I}_N) + (\boldsymbol{e}_i \otimes \boldsymbol{e}_i)'(\boldsymbol{I}_N \otimes \boldsymbol{P}_{(A0)}^{-1})\}\boldsymbol{U},$$

$i = 1, \ldots, N$; $j = 0, 1, 2$. Finally, the $(i, j)$ sub-block of the last block is equal to

$$[\boldsymbol{B}_{33t}]_{ij} = \mathsf{E}\boldsymbol{s}_t(\boldsymbol{\rho}_{Ai})\boldsymbol{s}_t'(\boldsymbol{\rho}_{Aj})$$
$$= \frac{1}{4}(t/T)^{i+j}\boldsymbol{U}'\boldsymbol{M}_A\boldsymbol{U},$$

$i, j = 0, 1, 2$, where

$$\boldsymbol{M}_A = \boldsymbol{P}_{(A0)}^{-1} \otimes \boldsymbol{P}_{(A0)}^{-1} + (\boldsymbol{P}_{(A0)}^{-1} \otimes \boldsymbol{I}_N)\boldsymbol{K}(\boldsymbol{P}_{(A0)}^{-1} \otimes \boldsymbol{I}_N).$$

**Proof.** See the appendix of Silvennoinen and Teräsvirta (2005) or Silvennoinen and Teräsvirta (2021). □

In order to define the test statistic, let $\boldsymbol{B}_{13\cdot j}$ be the $(i,j)$ blocks of $\boldsymbol{B}_{13}$ where $i \in \{1,\ldots,N\}$—that is,

$$\boldsymbol{B}_{13\cdot j} = \big[[\boldsymbol{B}'_{13}]_{j1},\ldots,[\boldsymbol{B}'_{13}]_{jN}\big], \quad j = 0,1,2$$

and define $\boldsymbol{B}_{23\cdot j}$ similarly. Partition the matrix $\boldsymbol{B}$ as follows:

$$\tilde{\boldsymbol{B}}_{11} = \begin{bmatrix} \boldsymbol{B}_{11} & \boldsymbol{B}_{12} & \boldsymbol{B}_{13\cdot 0} \\ \boldsymbol{B}'_{12} & \boldsymbol{B}_{22} & \boldsymbol{B}_{23\cdot 0} \\ \boldsymbol{B}'_{13\cdot 0} & \boldsymbol{B}'_{23\cdot 0} & [\boldsymbol{B}_{33}]_{00} \end{bmatrix},$$

$$\tilde{\boldsymbol{B}}_{12} = \begin{bmatrix} \boldsymbol{B}_{13\cdot 1} & \boldsymbol{B}_{13\cdot 2} \\ [\boldsymbol{B}_{33}]_{01} & [\boldsymbol{B}_{33}]_{02} \end{bmatrix}$$

and

$$\tilde{\boldsymbol{B}}_{33} = \begin{bmatrix} [\boldsymbol{B}_{33}]_{11} & [\boldsymbol{B}_{33}]_{12} \\ [\boldsymbol{B}'_{33}]_{12} & [\boldsymbol{B}_{33}]_{22} \end{bmatrix}.$$

Next, define

$$\widehat{\boldsymbol{x}}_{jt} = -\frac{1}{2}\Big(\frac{t}{T}\Big)^j \boldsymbol{U}'\big\{\mathrm{vec}(\boldsymbol{P}_{(A0)}^{-1}) - (\widehat{\boldsymbol{P}}_{(A0)}^{-1} \otimes \widehat{\boldsymbol{P}}_{(A0)}^{-1})\mathrm{vec}(\widehat{\boldsymbol{z}}_t \widehat{\boldsymbol{z}}'_t)\big\},$$

$j = 1,2$, where $\widehat{\boldsymbol{z}}_t$ and $\widehat{\boldsymbol{P}}_{(A0)}$ equal $\boldsymbol{z}_t$ and $\boldsymbol{P}_{(A0)}$ estimated under $\mathrm{H}_0$, respectively. The test statistic

$$LM_T = T\Big(\frac{1}{T}\sum_{t=1}^{T}\widehat{\boldsymbol{x}}'_{1t}, \frac{1}{T}\sum_{t=1}^{T}\widehat{\boldsymbol{x}}'_{2t}\Big)\big\{\tilde{\boldsymbol{B}}_{22} - \tilde{\boldsymbol{B}}'_{12}(\tilde{\boldsymbol{B}}^0_{11})^{-1}\tilde{\boldsymbol{B}}_{12}\big\}^{-1}\Big(\frac{1}{T}\sum_{t=1}^{T}\widehat{\boldsymbol{x}}'_{1t}, \frac{1}{T}\sum_{t=1}^{T}\widehat{\boldsymbol{x}}'_{2t}\Big)' \quad (A5)$$

has an asymptotic $\chi^2$-distribution with $N(N-1)$ degrees of freedom when $\mathrm{H}_0$ holds. To make the test statistic operational, the sub-blocks of the information matrix in (A5) have to be replaced by consistent plug-in estimators.

*Appendix A.4. Test for an Additional Transition in the Correlations*

The test statistic for an additional transition is constructed in the same way as in Appendix A.3, and the blocks related to the volatility components are identical. However, all blocks related to the correlation need modifications to include the parameters governing the time-varying correlation that exists under the null. This includes both the parameters in the correlation matrices under the null, and their corresponding transition parameters.

Let us define $\boldsymbol{x}_{hit} = h_{it}^{-1}\frac{\partial h_{it}}{\partial \boldsymbol{\theta}_{hi}}$, $\boldsymbol{x}_{git} = g_{it}^{-1}\frac{\partial g_{it}}{\partial \boldsymbol{\theta}_{gi}} + h_{it}^{-1}\frac{\partial h_{it}}{\partial \boldsymbol{\theta}_{gi}}$. Let us also partition the linearised correlation model as $\boldsymbol{P}_{At} = \boldsymbol{P}_{At0} + t/T\boldsymbol{P}_{(A1)} + (t/T)^2\boldsymbol{P}_{(A2)}$, where $\boldsymbol{P}_{At0}$ contains the time-varying correlation model under the null. When testing $L$ transitions against $L+1$ transitions, $\boldsymbol{P}_{At0}$ contains $L+1$ correlation matrices $\boldsymbol{P}_{(1)},\ldots,\boldsymbol{P}_{(L+1)}$ and $L$ transition functions $G_l(t/T,\gamma_l,c_l)$ (here, we assume $K_l = 1$ for simplicity), $l = 1,\ldots,L$. The information matrix is approximated by its consistent estimator

$$\widehat{\boldsymbol{B}} = T^{-1}\sum_{t=1}^{T}\mathsf{E}_{t-1}[\boldsymbol{s}_t(\boldsymbol{\theta}^0)\boldsymbol{s}_t(\boldsymbol{\theta}^0)'],$$

where $\boldsymbol{\theta}_0 = (\boldsymbol{\theta}_g, \boldsymbol{\theta}_h, \boldsymbol{\theta}_G, \boldsymbol{\theta}_{\rho A0}, \boldsymbol{\theta}_{\rho A1})'$, where $\boldsymbol{\theta}_G$ contains the transition parameters from the $L$ transitions that are present under the null, $\boldsymbol{\theta}_{\rho A0} = (\boldsymbol{\rho}_{(1)},\ldots,\boldsymbol{\rho}_{(L+1)})'$ and $\boldsymbol{\theta}_{\rho A1} = (\boldsymbol{\rho}_{(A1)}, \boldsymbol{\rho}_{(A2)})'$. From here on, the expressions are evaluated at the true parameter values under the null (we omit the additional superscripts of 0 to keep the notation simple).

With this notation, the $(i, j)$ sub-block of $\hat{B}_{11}$, $i \neq j$, equals

$$[\hat{B}_{11}]_{ij} = \frac{1}{4T} \sum_{t=1}^{T} x_{git} x'_{gjt} e'_i P^{-1}_{(At0)} e_j e'_i P_{(At0)} e_j.$$

When $i = j$,

$$[\hat{B}_{11}]_{ii} = \frac{1}{4T} \sum_{t=1}^{T} x_{git} x'_{git} (1 + e'_i P^{-1}_{(At0)} e_i).$$

Similarly, the $(i, j)$ sub-block of $\hat{B}_{22}$, $i \neq j$, is equal to

$$[\hat{B}_{22}]_{ij} = \frac{1}{4T} \sum_{t=1}^{T} x_{hit} x'_{hjt} e'_i P^{-1}_{(At0)} e_j e'_i P_{(At0)} e_j.$$

When $i = j$,

$$[\hat{B}_{22}]_{ii} = \frac{1}{4T} \sum_{t=1}^{T} x_{hit} x'_{hit} (1 + e'_i P^{-1}_{(At0)} e_i).$$

The $(i, j)$ sub-block of $\hat{B}_{12}$, $i \neq j$, equals

$$[\hat{B}_{12}]_{ij} = \frac{1}{4T} \sum_{t=1}^{T} x_{git} x'_{hjt} e'_i P^{-1}_{(At0)} e_j e'_i P_{(At0)} e_j.$$

When $i = j$,

$$[\hat{B}_{12}]_{ii} = \frac{1}{4T} \sum_{t=1}^{T} x_{git} x'_{hit} (1 + e'_i P^{-1}_{(At0)} e_i).$$

The next blocks deal with the transition parameters. Define $x_{Gt} = \frac{\partial \mathrm{vec} P'_{At}}{\partial \theta_G}$. The $l$th block of $\frac{\partial \mathrm{vec} P'_{At}}{\partial \theta_G}$ is

$$\left( \prod_{i=l+1}^{L} (1 - G_i) \right) \frac{\partial G_i}{\partial \theta_{G_i}} \mathrm{vec}(P_{(l+1)} - P_t^{(l-1)})'$$

using the recursion in (8). The block $\hat{B}_{33}$ is equal to

$$\hat{B}_{33} = \frac{1}{4T} \sum_{t=1}^{T} x_{Gt} M_A x'_{Gt}.$$

The $i$th sub-block of $\hat{B}_{13}$ equals

$$[\hat{B}_{13}]_i = \frac{1}{4T} \sum_{t=1}^{T} x_{git} (e'_i P^{-1}_{(At0)} \otimes e'_i + e'_i \otimes e'_i P^{-1}_{(At0)}) x'_{Gt}$$

and the $i$th sub-block of $\hat{B}_{23}$ equals

$$[\hat{B}_{23}]_i = \frac{1}{4T} \sum_{t=1}^{T} x_{hit} (e'_i P^{-1}_{(At0)} \otimes e'_i + e'_i \otimes e'_i P^{-1}_{(At0)}) x'_{Gt}.$$

Next, we will consider the blocks related to the correlations. The matrix $\partial \text{vec}(\boldsymbol{P}_{At})'/\partial \boldsymbol{\theta}_{\rho_{A0}}$ equals

$$\frac{\partial \text{vec}(\boldsymbol{P}_{At})'}{\partial \boldsymbol{\theta}_{\rho_{A0}}} = vs. \otimes \boldsymbol{U}' = \begin{bmatrix} \prod_{l=1}^{L}(1 - G_{lt}) \\ G_{1t} \prod_{l=2}^{L}(1 - G_{lt}) \\ G_{2t} \prod_{l=3}^{L}(1 - G_{lt}) \\ \cdots \\ G_{l-1,t}(1 - G_{Lt}) \\ G_{Lt} \end{bmatrix} \otimes \boldsymbol{U}'$$

and the matrix $\partial \text{vec}(\boldsymbol{P}_{At})'/\partial \boldsymbol{\theta}_{\rho_{A1}}$ equals

$$\frac{\partial \text{vec}(\boldsymbol{P}_{At})'}{\partial \boldsymbol{\rho}_{A1}} = \begin{bmatrix} t/T \\ (t/T)^2 \end{bmatrix} \otimes \boldsymbol{U}'.$$

The block $\hat{\boldsymbol{B}}_{44}$ is equal to

$$\hat{\boldsymbol{B}}_{44} = \frac{1}{4T} \sum_{t=1}^{T} v_t v_t' \otimes \boldsymbol{U}' \boldsymbol{M}_A \boldsymbol{U}$$

and the $(i, j)$ sub-block of $\hat{\boldsymbol{B}}_{55}$ is equal to

$$[\hat{\boldsymbol{B}}_{55}]_{ij} = \frac{1}{4T} \sum_{t=1}^{T} (t/T)^{i+j} \boldsymbol{U}' \boldsymbol{M}_A \boldsymbol{U}$$

for $i, j = 1, 2$. The $i$th sub-block of $\hat{\boldsymbol{B}}_{14}$ is equal to

$$[\hat{\boldsymbol{B}}_{14}]_i = \frac{1}{4T} \sum_{t=1}^{T} x_{git}(e_i' \boldsymbol{P}_{(At0)}^{-1} \otimes e_i' + e_i' \otimes e_i' \boldsymbol{P}_{(At0)}^{-1})(v_t' \otimes \boldsymbol{U})$$

and the $(i, j)$ sub-block of $\hat{\boldsymbol{B}}_{15}$ is equal to

$$[\hat{\boldsymbol{B}}_{15}]_{ij} = \frac{1}{4T} \sum_{t=1}^{T} (t/T)^j x_{git}(e_i' \boldsymbol{P}_{(At0)}^{-1} \otimes e_i' + e_i' \otimes e_i' \boldsymbol{P}_{(At0)}^{-1}) \boldsymbol{U},$$

$j = 1, 2$. The corresponding sub-blocks of $\hat{\boldsymbol{B}}_{24}$ and $\hat{\boldsymbol{B}}_{25}$ are

$$[\hat{\boldsymbol{B}}_{24}]_i = \frac{1}{4T} \sum_{t=1}^{T} x_{hit}(e_i' \boldsymbol{P}_{(At0)}^{-1} \otimes e_i' + e_i' \otimes e_i' \boldsymbol{P}_{(At0)}^{-1})(v_t' \otimes \boldsymbol{U})$$

and the $(i, j)$ sub-block of $\hat{\boldsymbol{B}}_{15}$ is equal to

$$[\hat{\boldsymbol{B}}_{25}]_{ij} = \frac{1}{4T} \sum_{t=1}^{T} (t/T)^j x_{hit}(e_i' \boldsymbol{P}_{(At0)}^{-1} \otimes e_i' + e_i' \otimes e_i' \boldsymbol{P}_{(At0)}^{-1}) \boldsymbol{U},$$

$j = 1, 2$. The block $\hat{\boldsymbol{B}}_{34}$ equals

$$\hat{\boldsymbol{B}}_{34} = \frac{1}{4T} \sum_{t=1}^{T} x_{Gt} \boldsymbol{M}_A(v' \otimes \boldsymbol{U}).$$

The $i$th sub-block of $\hat{\boldsymbol{B}}_{35}$ equals

$$[\hat{\boldsymbol{B}}_{35}]_i = \frac{1}{4T} \sum_{t=1}^{T} (t/T)^i x_{Gt} \boldsymbol{M}_A \boldsymbol{U},$$

$i = 1, 2$ Finally, the $i$th sub-block of $\hat{\boldsymbol{B}}_{45}$ is equal to

$$[\hat{\boldsymbol{B}}_{45}]_i = \frac{1}{4T} \sum_{t=1}^{T} (t/T)^i (vs. \otimes \boldsymbol{U}') \boldsymbol{M}_A \boldsymbol{U}.$$

Next, define

$$\widehat{\boldsymbol{x}}_{jt} = -\frac{1}{2} (\frac{t}{T})^j \boldsymbol{U}' \{ \text{vec}(\boldsymbol{P}_{(A0t)}^{-1}) - (\hat{\boldsymbol{P}}_{(A0t)}^{-1} \otimes \hat{\boldsymbol{P}}_{(A0t)}^{-1}) \text{vec}(\hat{\boldsymbol{z}}_t \hat{\boldsymbol{z}}_t') \},$$

$j = 1, 2$, where $\hat{\boldsymbol{z}}_t$ and $\hat{\boldsymbol{P}}_{(A0t)}$ equal $\boldsymbol{z}_t$ and $\boldsymbol{P}_{(A0t)}$ estimated under $H_0$, respectively. The test statistic

$$LM_T = T(\frac{1}{T} \sum_{t=1}^{T} \hat{\boldsymbol{x}}_{1t}', \frac{1}{T} \sum_{t=1}^{T} \hat{\boldsymbol{x}}_{2t}') [\hat{\boldsymbol{B}}^{-1}]_{SW} (\frac{1}{T} \sum_{t=1}^{T} \hat{\boldsymbol{x}}_{1t}', \frac{1}{T} \sum_{t=1}^{T} \hat{\boldsymbol{x}}_{2t}')',$$

where $[\hat{\boldsymbol{B}}^{-1}]_{SW}$ is the $N(N-1) \times N(N-1)$ block in the south-west corner of the inverse of $\hat{\boldsymbol{B}}$. As the matrix $\hat{\boldsymbol{B}}$ can have a large dimension, its inverse could be obtained by using block inversion methods, perhaps applying them recursively. The test statistic has an asymptotic $\chi^2$-distribution with $N(N-1)$ degrees of freedom when $H_0$ holds.

## Appendix B. Simulations of Test Statistics

### Appendix B.1. Tests of GARCH Equations

The test for slow moving baseline volatility has a statistic whose distribution is sensitive to the high frequency, GARCH, volatility. For this reason, one cannot use the asymptotic distribution, rather the distribution must be generated via simulation. Further, Silvennoinen and Teräsvirta (2016) showed that the size of the test was distorted if the GARCH parameterisation deviates from the true one. For this reason, a few alternative approaches to estimate the GARCH parameters, and especially the persistence, have been investigated. It should be noted that estimating GARCH without taking the nonstationarity into account will yield overestimated persistence, thereby, impacting the null distribution of the test statistic and thus rendering the test outcomes unreliable. These estimates are given in Table A1.

The baseline volatility may be very different in different series. Therefore, one should not ignore visual inspection of the returns nor rely on general rules of thumb. If there are sufficiently long sections of data where the general level of volatility remains constant, it is advisable to estimate the GARCH parameters over such subsample. In the present case, there are a couple of relatively constant volatility sections—for example, one from November 2003 until October 2007.

The parameter estimates for that calm subperiod are in Table A1. Comparison with the estimates from the entire period GARCH model makes it clear that the neglected nonstationarity has biased the estimates, resulting in high persistence and kurtosis. As the data set has a sufficiently long span of GARCH-type clustering without (visually) significant movement in the general baseline level, relevant estimates are obtained by using that subsample only.

Another approach consists of estimating the GARCH equation over a rolling window such that the intercept is time-varying, targeting the unconditional volatility over each window, while the other parameters are assumed to be constant over the entire sample period and estimated in the usual way. The choice of the window length should consider the general recommendations regarding the sample size when attempting GARCH estimation. Too long a window will be impacted by the slowly changing baseline volatility level, whereas too short a window will yield very uncertain GARCH estimates.

To investigate the properties of this approach, we ran a simulation experiment with a few different baseline volatilities. The window widths varied from 250 to 1000 observations. Figures A1–A3 depict the distributions of the GARCH estimates and the derived persistence and kurtosis measures, as explained in He and Teräsvirta (1999), for a selection of baseline

volatilities and window widths. Based on these experiments, we concluded that a window width of 400 observations yields sufficiently robust results for our application.

The resulting GARCH estimates are reported in Table A1, and they are quite similar to the ones obtained for the aforementioned calm period. This can be interpreted as support for the rolling window method, particularly in situations where visual inspection of data does not reveal a sufficiently long period of constant unconditional volatility.

Overall, it is clear that using simply the GARCH estimates from the entire sample to calibrate the null distribution of the test statistic for the specification of the deterministic component of the volatility is not recommended. For comparison, Table A1 also reports the GARCH estimates from a TV-GARCH model where the TV specification has been completed. The estimated persistence is higher than the ones obtained from the calm period or rolling window variance targeting method, however, as discussed in Silvennoinen and Teräsvirta (2016), underestimation of persistence has a less severe impact on the performance of the TV specification test than does overestimation.

**Table A1.** Specification stage for the deterministic component in volatilities of each of the four banks. $\tilde{\alpha}$ and $\tilde{\beta}$ are the initial estimates used for calibrating the test statistic distribution. The rolling window method allows the GARCH intercept to adjust to target the unconditional variance in a window of size 400. The 'calm period' selects the continuous period from November 2003 to October 2007, which has very little visible variation in the baseline volatility. For comparison, the GARCH estimates from the entire sample period are reported along with the final estimates from the TV-GARCH model.

|  |  | $\tilde{\alpha}$ | $\tilde{\beta}$ | Persistence | Kurtosis |
|---|---|---|---|---|---|
| Rolling window 400 | ANZ | 0.090 | 0.836 | 0.926 | 3.38 |
|  | CBA | 0.087 | 0.850 | 0.937 | 3.43 |
|  | NAB | 0.095 | 0.817 | 0.912 | 3.36 |
|  | WBC | 0.085 | 0.858 | 0.943 | 3.45 |
| Calm period | ANZ | 0.073 | 0.852 | 0.925 | 3.24 |
|  | CBA | 0.081 | 0.842 | 0.923 | 3.29 |
|  | NAB | 0.066 | 0.829 | 0.896 | 3.14 |
|  | WBC | 0.091 | 0.806 | 0.897 | 3.28 |
| Entire period GARCH only | ANZ | 0.065 | 0.927 | 0.992 | 6.40 |
|  | CBA | 0.089 | 0.890 | 0.979 | 4.83 |
|  | NAB | 0.104 | 0.867 | 0.971 | 4.85 |
|  | WBC | 0.075 | 0.911 | 0.986 | 5.08 |
| Entire period TV-GARCH | ANZ | 0.078 | 0.880 | 0.957 | 3.50 |
|  | CBA | 0.091 | 0.860 | 0.950 | 3.61 |
|  | NAB | 0.107 | 0.825 | 0.931 | 3.62 |
|  | WBC | 0.084 | 0.878 | 0.962 | 3.70 |

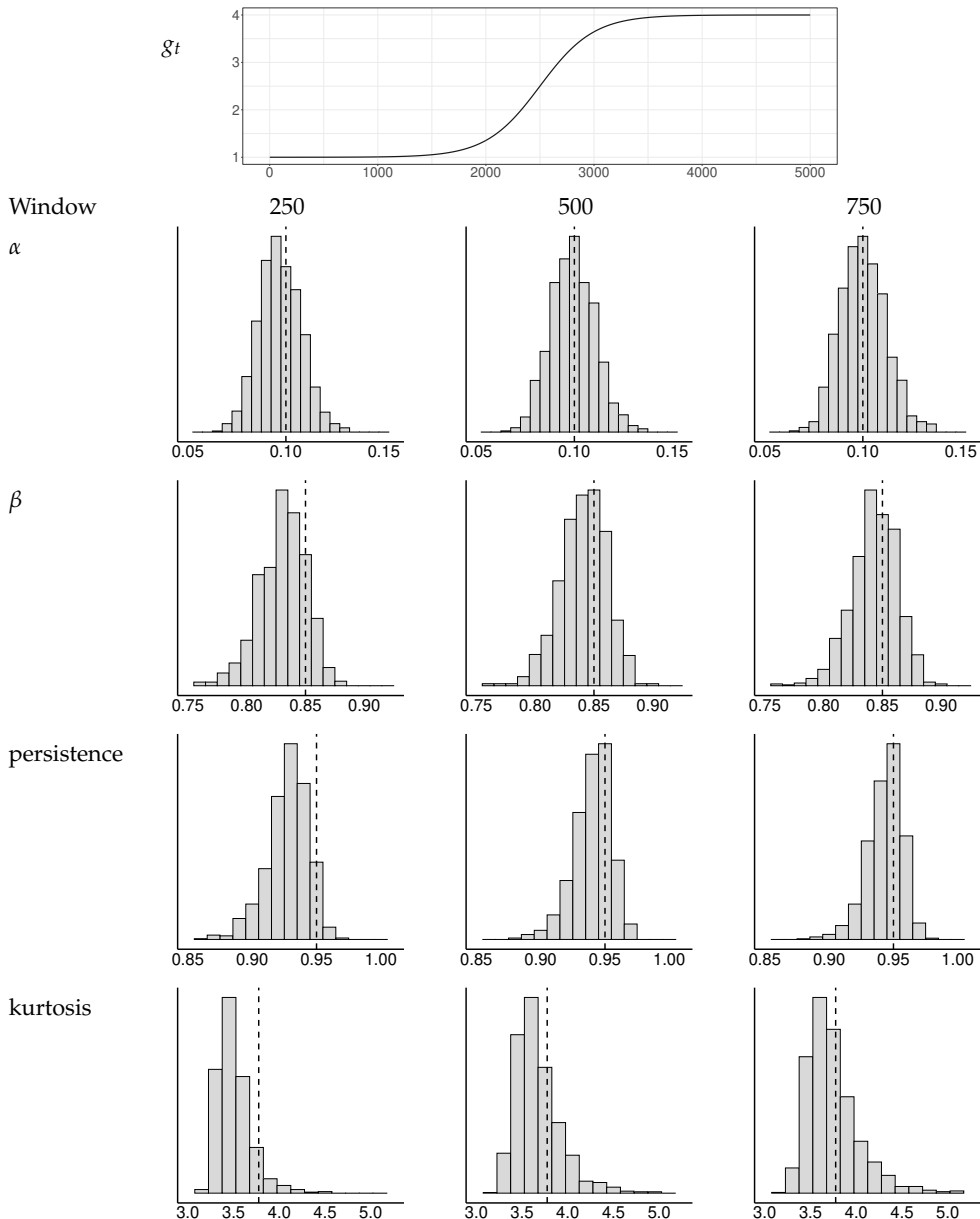

**Figure A1.** Simulated distributions of GARCH estimates and implied persistence and kurtosis measures for a selection of window widths. The baseline $g_t$ has a single transition. The dotted vertical lines indicate the true values of the parameters $\alpha$, $\beta$, persistence, and kurtosis.

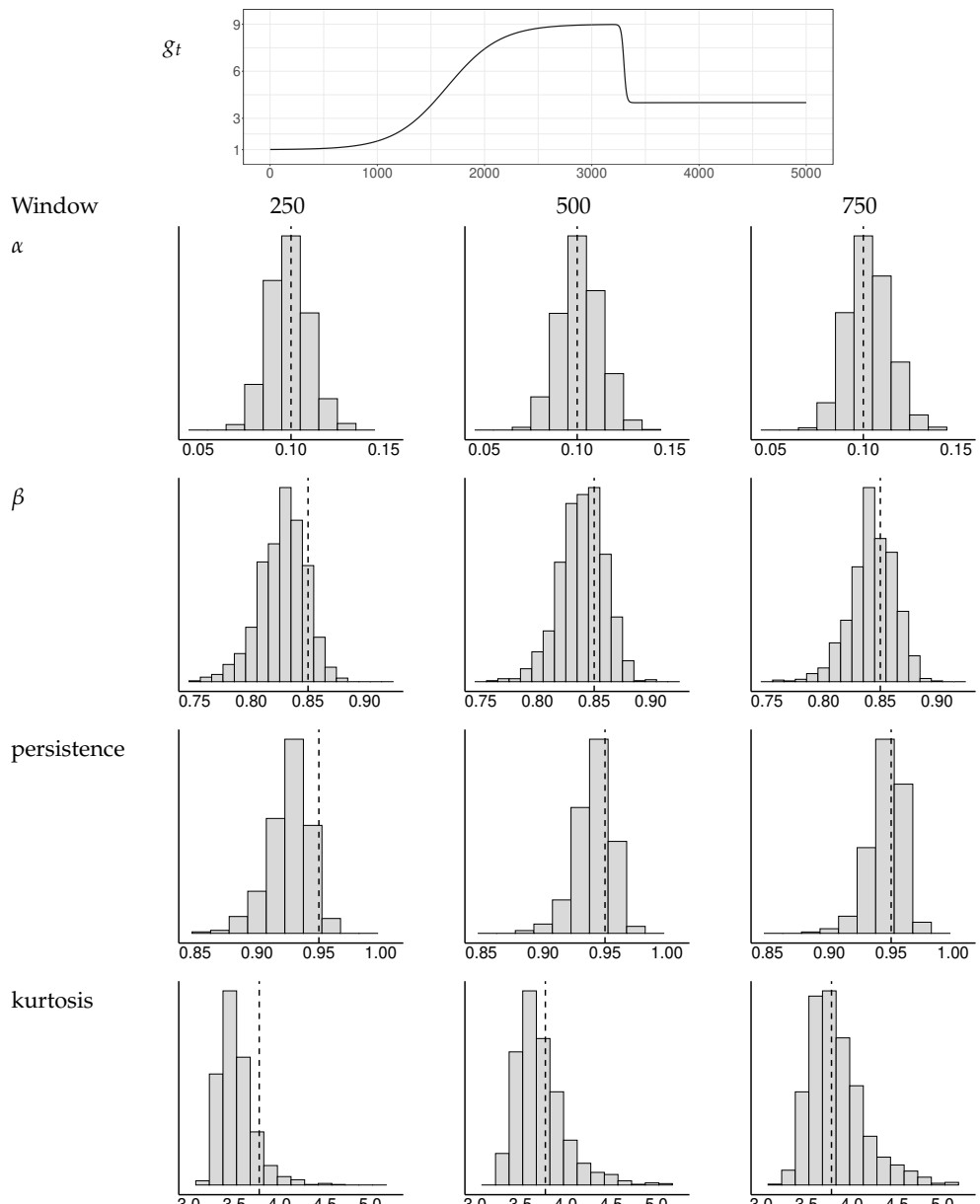

**Figure A2.** Simulated distributions of GARCH estimates and implied persistence and kurtosis measures for a selection of window widths. The baseline $g_t$ has an asymmetric double transition. The dotted vertical lines indicate the true values of the parameters $\alpha$, $\beta$, persistence, and kurtosis.

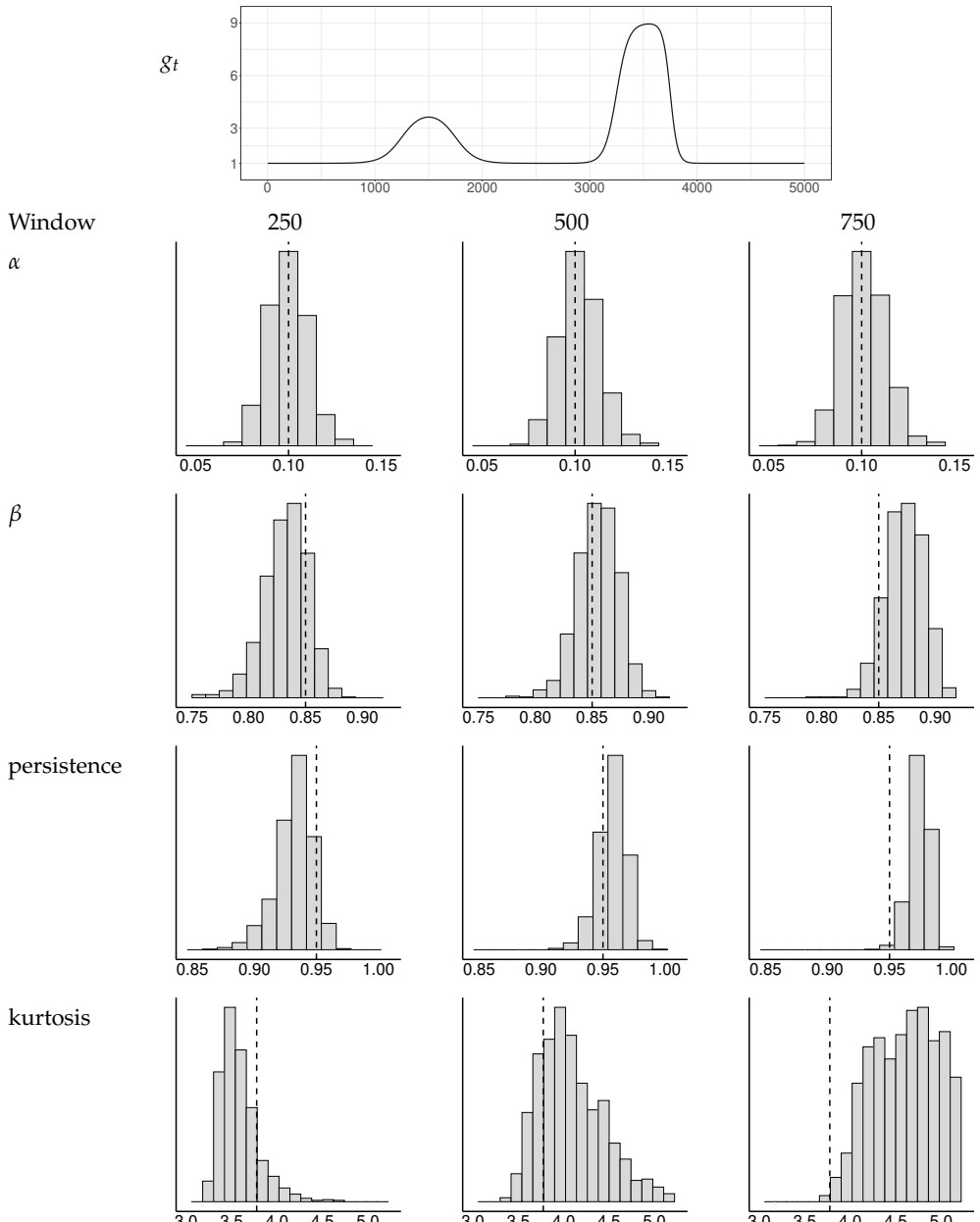

**Figure A3.** Simulated distributions of GARCH estimates and implied persistence and kurtosis measures for a selection of window widths. The baseline $g_t$ has two double transitions. The dotted vertical lines indicate the true values of the parameters $\alpha$, $\beta$, persistence, and kurtosis.

### *Appendix B.2. Evaluation Tests of GARCH Equations*

The fact that the evaluation tests discussed in Appendix A.2 are applied to the pre-filtered data $\hat{P}_t^{-1/2}\varepsilon_t$ is known to potentially alter the distribution of the test statistic. In this section, we present simulation results that show that the size of the tests remains practically unchanged, rendering the tests applicable in the proposed way.

The simulation uses 2000 observations on a bivariate TVGARCH model parametrised as $h_t = 0.10 + 0.05\varepsilon_{t-1}^2/g_{t-1} + 0.85h_{t-1}$, $g_t = 1 + 3(1 + \exp\{-e^3(t/T - 0.5)\})^{-1}$. These are coupled with a CCC model with $\rho = 0.5$, and then with an STCC model parametrised as $\rho_{(1)} = 0.3$, $\rho_{(2)} = 0.7$, $G_t = (1 + \exp\{-e^{2.5}(t/T - 0.5)\})^{-1}$. The noise terms are iid standard normal. Two estimation procedures were used, a two-step and a multi-step one.

First step  The individual TVGARCH models are estimated, assuming the series are uncorrelated.

Second step    Estimate the correlation model conditional on the volatility model estimates from the previous step. Then, estimate the TVGARCH models conditional on the correlation estimates.

The misspecification tests are then calculated using the TVGARCH estimates from the second step, and the data is pre-filtered with the correlation estimates from the second step. The multi-step continues repeating the procedure of the second step until no further improvements are achieved.

**Table A2.** Size simulation for the three types of misspecification tests in Amado and Teräsvirta (2017). 2000 replications. $T = 2000$, $N = 2$. MS1: $g_t$ additively misspecified, alternative linearised with a first-order term only; MS2-a: GARCH(1,1) vs. GARCH(1,2); MS2-b: GARCH(1,1) vs. GARCH(2,1); MS3: test for remaining ARCH, lag 1.

|  |  | Standard | | | Robust | | |
|---|---|---|---|---|---|---|---|
|  |  | **10%** | **5%** | **1%** | **10%** | **5%** | **1%** |
| CCC two-step | MS1 | 0.146 | 0.085 | 0.020 | 0.132 | 0.074 | 0.016 |
|  | MS2-a | 0.122 | 0.064 | 0.012 | 0.101 | 0.048 | 0.013 |
|  | MS2-b | 0.143 | 0.080 | 0.017 | 0.108 | 0.051 | 0.008 |
|  | MS3 | 0.125 | 0.061 | 0.010 | 0.104 | 0.054 | 0.010 |
| STCC two-step | MS1 | 0.134 | 0.074 | 0.023 | 0.121 | 0.055 | 0.015 |
|  | MS2-a | 0.123 | 0.059 | 0.015 | 0.101 | 0.045 | 0.013 |
|  | MS2-b | 0.122 | 0.062 | 0.019 | 0.087 | 0.044 | 0.010 |
|  | MS3 | 0.115 | 0.058 | 0.015 | 0.100 | 0.050 | 0.011 |
| CCC multi-step | MS1 | 0.145 | 0.083 | 0.022 | 0.133 | 0.073 | 0.014 |
|  | MS2-a | 0.116 | 0.062 | 0.015 | 0.097 | 0.052 | 0.009 |
|  | MS2-b | 0.133 | 0.069 | 0.018 | 0.100 | 0.046 | 0.010 |
|  | MS3 | 0.120 | 0.062 | 0.016 | 0.107 | 0.060 | 0.014 |
| STCC multi-step | MS1 | 0.147 | 0.084 | 0.023 | 0.135 | 0.068 | 0.012 |
|  | MS2-a | 0.130 | 0.059 | 0.011 | 0.103 | 0.046 | 0.006 |
|  | MS2-b | 0.120 | 0.067 | 0.016 | 0.090 | 0.039 | 0.005 |
|  | MS3 | 0.112 | 0.055 | 0.012 | 0.104 | 0.047 | 0.009 |

From Table A2, it is evident that the standard form of the tests is slightly oversized. The robust version of the tests, on the other hand, seems to behave well, and there is no need for any adjustments of the test statistics or their distributions. Therefore, the procedure of removing the correlations between the series prior to applying the evaluation tests can be recommended.

*Appendix B.3. Tests of Correlations*

The simulation experiment investigates the size of the test in an environment where the multivariate model is correctly specified. The number of data series considered in the system is $N = 2, 5, 10, 20$. The length varies from $T = 25$ for the bivariate systems, which is relevant for time series systems in macro applications, up to $T = 1000$, which, in turn, is considered to be a fairly small sample size for high frequency returns data. The length of the time series places a constraint on the dimension of the model—that is, the parametric alternative is only feasible if the number of parameters remains comfortably below the amount of available data points. We simulated the test by both assuming that $\boldsymbol{D}_t \equiv \boldsymbol{I}_N$ and that there is conditional heteroskedasticity in the model: $\boldsymbol{D}_t \neq \boldsymbol{I}_N$.

When $\boldsymbol{D}_t \equiv \boldsymbol{I}_N$, we found that the results were fairly independent of the structure of the correlations. We used both equicorrelation and Toeplitz matrices in our simulations, and the results remained the same. Table A3 contains the results of a simulation in which $\boldsymbol{D}_t \equiv \boldsymbol{I}_N$, and the $N \times N$ correlation matrix $\boldsymbol{P} = [\rho_{ij}]$ is an equicorrelated one with weak ($\rho = 1/3$) and moderately strong ($\rho = 2/3$) correlation. The table also reports the results from using a Toeplitz correlation matrix such that $[\rho_{ij}] = \rho^{|i-j|}$, $i, j = 1, \ldots, N$ with $\rho = 0.5$ representing moderate to weak correlation and $\rho = 0.9$ representing strong to moderate

correlation. It is seen that the empirical size of the test is rather close to the nominal one already when $N = 2$ and $T = 100$. The size holds up across the various correlation patterns.

**Table A3.** Size-study: Test of constant correlations. Data are generated as an MTV-CCC with an equicorrelation coefficient of 0.33 (CEC33) and 0.67 (CEC67) and a Toeplitz structure with a correlation coefficient of 0.5 (CTC50) and 0.9 (CTC90). Tests are based on the first-order polynomial approximation. A total of 5000 replications.

| | | CEC33 | | | CEC67 | | | CTC50 | | | CTC90 | | |
|---|---|---|---|---|---|---|---|---|---|---|---|---|---|
| N | T | 1% | 5% | 10% | 1% | 5% | 10% | 1% | 5% | 10% | 1% | 5% | 10% |
| 2 | 25 | 0.023 | 0.076 | 0.132 | 0.022 | 0.069 | 0.128 | 0.024 | 0.074 | 0.130 | 0.022 | 0.070 | 0.126 |
| | 50 | 0.015 | 0.063 | 0.116 | 0.016 | 0.064 | 0.115 | 0.016 | 0.064 | 0.115 | 0.015 | 0.062 | 0.109 |
| | 100 | 0.011 | 0.056 | 0.104 | 0.010 | 0.054 | 0.102 | 0.011 | 0.056 | 0.103 | 0.010 | 0.051 | 0.101 |
| | 250 | 0.012 | 0.055 | 0.108 | 0.010 | 0.054 | 0.107 | 0.011 | 0.055 | 0.106 | 0.009 | 0.053 | 0.108 |
| | 500 | 0.010 | 0.051 | 0.097 | 0.009 | 0.049 | 0.097 | 0.010 | 0.050 | 0.096 | 0.009 | 0.050 | 0.094 |
| | 1000 | 0.010 | 0.048 | 0.099 | 0.010 | 0.048 | 0.095 | 0.010 | 0.046 | 0.097 | 0.010 | 0.049 | 0.092 |
| 5 | 100 | 0.011 | 0.054 | 0.112 | 0.011 | 0.053 | 0.110 | 0.011 | 0.056 | 0.112 | 0.011 | 0.053 | 0.111 |
| | 250 | 0.014 | 0.054 | 0.099 | 0.012 | 0.051 | 0.099 | 0.013 | 0.053 | 0.100 | 0.012 | 0.051 | 0.101 |
| | 500 | 0.010 | 0.050 | 0.104 | 0.010 | 0.053 | 0.106 | 0.009 | 0.052 | 0.101 | 0.010 | 0.054 | 0.105 |
| | 1000 | 0.010 | 0.056 | 0.102 | 0.010 | 0.052 | 0.103 | 0.009 | 0.053 | 0.100 | 0.008 | 0.053 | 0.103 |
| 10 | 250 | 0.013 | 0.055 | 0.112 | 0.013 | 0.057 | 0.112 | 0.013 | 0.057 | 0.110 | 0.012 | 0.054 | 0.115 |
| | 500 | 0.009 | 0.049 | 0.101 | 0.010 | 0.049 | 0.104 | 0.008 | 0.053 | 0.103 | 0.010 | 0.050 | 0.103 |
| | 1000 | 0.011 | 0.052 | 0.102 | 0.011 | 0.054 | 0.105 | 0.011 | 0.053 | 0.099 | 0.012 | 0.056 | 0.103 |
| 20 | 1000 | 0.012 | 0.056 | 0.106 | 0.012 | 0.057 | 0.106 | 0.013 | 0.056 | 0.103 | 0.012 | 0.056 | 0.107 |

We next turn to the case $D_t \neq I_N$. Tables A4 and A5 contain results of size simulations where the sensitivity of the test is examined against combinations for the GARCH persistence and kurtosis as well as a selection of strengths of correlations (the equicorrelated and Toepliz ones described above). The test is generally well-sized.

**Table A4.** Size-study: Test of constant correlations. Data are generated as an MTV-GARCH-CEC with persistence of 0.95 and 0.97, kurtosis of 4 and 6, and an equicorrelation coefficient of 0.33 and 0.67. Tests are based on the first-order polynomial approximation. A total of 2500 replications.

| | | | CEC33 | | | | | | CEC67 | | | | | |
|---|---|---|---|---|---|---|---|---|---|---|---|---|---|---|
| | | | kurtosis = 4 | | | kurtosis = 6 | | | kurtosis = 4 | | | kurtosis = 6 | | |
| Persistence | N | T | 1% | 5% | 10% | 1% | 5% | 10% | 1% | 5% | 10% | 1% | 5% | 10% |
| 0.95 | 2 | 500 | 0.012 | 0.056 | 0.108 | 0.016 | 0.056 | 0.106 | 0.016 | 0.070 | 0.122 | 0.016 | 0.092 | 0.122 |
| | 2 | 1000 | 0.009 | 0.044 | 0.103 | 0.009 | 0.042 | 0.097 | 0.011 | 0.045 | 0.093 | 0.009 | 0.044 | 0.097 |
| | 2 | 2000 | 0.008 | 0.042 | 0.094 | 0.007 | 0.042 | 0.090 | 0.010 | 0.052 | 0.099 | 0.009 | 0.046 | 0.092 |
| | 5 | 500 | 0.006 | 0.062 | 0.118 | 0.006 | 0.070 | 0.114 | 0.018 | 0.076 | 0.140 | 0.018 | 0.082 | 0.146 |
| | 5 | 1000 | 0.016 | 0.060 | 0.119 | 0.016 | 0.061 | 0.112 | 0.016 | 0.059 | 0.115 | 0.018 | 0.060 | 0.112 |
| | 5 | 2000 | 0.010 | 0.058 | 0.108 | 0.008 | 0.051 | 0.102 | 0.016 | 0.060 | 0.116 | 0.010 | 0.052 | 0.098 |
| | 10 | 500 | 0.016 | 0.058 | 0.118 | 0.020 | 0.064 | 0.114 | 0.020 | 0.068 | 0.116 | 0.024 | 0.080 | 0.128 |
| | 10 | 1000 | 0.018 | 0.053 | 0.104 | 0.015 | 0.051 | 0.101 | 0.014 | 0.061 | 0.111 | 0.017 | 0.063 | 0.110 |
| | 10 | 2000 | 0.014 | 0.072 | 0.126 | 0.012 | 0.060 | 0.112 | 0.018 | 0.082 | 0.142 | 0.013 | 0.062 | 0.118 |
| 0.97 | 2 | 500 | 0.010 | 0.056 | 0.114 | 0.012 | 0.054 | 0.118 | 0.020 | 0.072 | 0.114 | 0.014 | 0.068 | 0.120 |
| | 2 | 1000 | 0.011 | 0.043 | 0.102 | 0.011 | 0.044 | 0.103 | 0.012 | 0.047 | 0.107 | 0.013 | 0.048 | 0.103 |
| | 2 | 2000 | 0.009 | 0.046 | 0.094 | 0.007 | 0.042 | 0.089 | 0.010 | 0.056 | 0.108 | 0.012 | 0.050 | 0.093 |
| | 5 | 500 | 0.004 | 0.066 | 0.124 | 0.012 | 0.056 | 0.104 | 0.012 | 0.088 | 0.152 | 0.018 | 0.086 | 0.164 |
| | 5 | 1000 | 0.015 | 0.063 | 0.113 | 0.014 | 0.067 | 0.114 | 0.018 | 0.063 | 0.121 | 0.019 | 0.060 | 0.125 |
| | 5 | 2000 | 0.010 | 0.060 | 0.110 | 0.008 | 0.050 | 0.100 | 0.015 | 0.060 | 0.118 | 0.012 | 0.050 | 0.101 |
| | 10 | 500 | 0.012 | 0.062 | 0.108 | 0.016 | 0.070 | 0.112 | 0.016 | 0.072 | 0.112 | 0.022 | 0.086 | 0.148 |
| | 10 | 1000 | 0.016 | 0.053 | 0.100 | 0.015 | 0.056 | 0.107 | 0.015 | 0.063 | 0.113 | 0.018 | 0.057 | 0.110 |
| | 10 | 2000 | 0.015 | 0.074 | 0.132 | 0.014 | 0.058 | 0.108 | 0.016 | 0.088 | 0.142 | 0.010 | 0.063 | 0.112 |

**Table A5.** Size-study: Test of constant correlations. Data are generated as an MTV-GARCH-CTC with persistence of 0.95 and 0.97, kurtosis of 4 and 6, and a correlation matrix with a Toeplitz structure with a correlation coefficient of 0.5 and 0.9. Tests are based on the first-order polynomial approximation. A total of 2500 replications.

| | | | CTC50 | | | | | | CTC90 | | | | | |
| | | | kurtosis = 4 | | | kurtosis = 6 | | | kurtosis = 4 | | | kurtosis = 6 | | |
| Persistence | N | T | 1% | 5% | 10% | 1% | 5% | 10% | 1% | 5% | 10% | 1% | 5% | 10% |
|---|---|---|---|---|---|---|---|---|---|---|---|---|---|---|
| 0.95 | 2 | 500 | 0.010 | 0.064 | 0.102 | 0.010 | 0.070 | 0.106 | 0.018 | 0.094 | 0.136 | 0.026 | 0.088 | 0.146 |
| | 2 | 1000 | 0.009 | 0.041 | 0.097 | 0.011 | 0.042 | 0.103 | 0.014 | 0.053 | 0.096 | 0.020 | 0.062 | 0.104 |
| | 2 | 2000 | 0.008 | 0.044 | 0.096 | 0.009 | 0.044 | 0.090 | 0.017 | 0.066 | 0.120 | 0.014 | 0.048 | 0.098 |
| | 5 | 500 | 0.006 | 0.062 | 0.118 | 0.010 | 0.058 | 0.114 | 0.020 | 0.120 | 0.212 | 0.050 | 0.134 | 0.210 |
| | 5 | 1000 | 0.014 | 0.060 | 0.112 | 0.018 | 0.064 | 0.113 | 0.027 | 0.076 | 0.134 | 0.034 | 0.093 | 0.144 |
| | 5 | 2000 | 0.011 | 0.057 | 0.110 | 0.008 | 0.052 | 0.105 | 0.020 | 0.075 | 0.142 | 0.018 | 0.058 | 0.110 |
| | 10 | 500 | 0.012 | 0.070 | 0.120 | 0.016 | 0.080 | 0.128 | 0.040 | 0.114 | 0.172 | 0.078 | 0.150 | 0.230 |
| | 10 | 1000 | 0.012 | 0.049 | 0.100 | 0.013 | 0.051 | 0.102 | 0.019 | 0.078 | 0.127 | 0.032 | 0.089 | 0.147 |
| | 10 | 2000 | 0.019 | 0.072 | 0.127 | 0.014 | 0.059 | 0.111 | 0.033 | 0.110 | 0.178 | 0.018 | 0.077 | 0.140 |
| 0.97 | 2 | 500 | 0.014 | 0.066 | 0.114 | 0.018 | 0.068 | 0.116 | 0.016 | 0.082 | 0.134 | 0.030 | 0.104 | 0.164 |
| | 2 | 1000 | 0.009 | 0.044 | 0.101 | 0.008 | 0.042 | 0.099 | 0.016 | 0.051 | 0.112 | 0.022 | 0.063 | 0.119 |
| | 2 | 2000 | 0.010 | 0.050 | 0.102 | 0.009 | 0.044 | 0.092 | 0.024 | 0.070 | 0.120 | 0.015 | 0.052 | 0.100 |
| | 5 | 500 | 0.014 | 0.056 | 0.128 | 0.008 | 0.074 | 0.130 | 0.024 | 0.134 | 0.208 | 0.052 | 0.160 | 0.256 |
| | 5 | 1000 | 0.013 | 0.059 | 0.112 | 0.016 | 0.066 | 0.123 | 0.022 | 0.082 | 0.157 | 0.037 | 0.102 | 0.164 |
| | 5 | 2000 | 0.014 | 0.062 | 0.112 | 0.010 | 0.052 | 0.101 | 0.028 | 0.086 | 0.145 | 0.020 | 0.066 | 0.116 |
| | 10 | 500 | 0.018 | 0.080 | 0.128 | 0.022 | 0.088 | 0.130 | 0.040 | 0.114 | 0.172 | 0.100 | 0.188 | 0.278 |
| | 10 | 1000 | 0.012 | 0.054 | 0.105 | 0.013 | 0.054 | 0.107 | 0.019 | 0.078 | 0.127 | 0.030 | 0.104 | 0.181 |
| | 10 | 2000 | 0.016 | 0.072 | 0.132 | 0.016 | 0.062 | 0.110 | 0.033 | 0.110 | 0.178 | 0.026 | 0.089 | 0.150 |

However, an interesting aspect is that there is slight oversizing when kurtosis decreases (which means shifting the relative weight from $\alpha$ to $\beta$ in the GARCH equation, while keeping the persistence constant). A change in persistence does not seem to affect the size of the test. As the dimension of the system increases, the test does not perform equally well. Increasing the sample size does not seem to be able to counteract this (the simulations use $T = 500, 1000, 2000$).

In yet another simulation (results not reported here), we considered the effects of misspecifying the conditional heteroskedasticity on the correlation test. More specifically, when $D_t \neq I_N$ but conditional heteroskedasticity is ignored, the test is, as may be expected, heavily oversized. The obvious conclusion is that the constancy of correlations can only be tested after specifying and estimating both $D_t$ and $S_t$.

**Appendix C. Details of Maximisation by Parts**

This appendix describing the outlines of the estimation algorithm derives from Silvennoinen and Teräsvirta (2021). The estimation proceeds as follows.

1. Assume $\ln h_{it}(\boldsymbol{\theta}_{hi}, \boldsymbol{\theta}_{gi}) = 0$, $i = 1, \ldots, N$, and estimate parameters $\boldsymbol{\theta}_g = (\boldsymbol{\theta}_{g1}, \ldots, \boldsymbol{\theta}_{gN})'$, $i = 1, \ldots, N$, equation by equation, assuming $P_t(\boldsymbol{\theta}_\rho) = I_N$. Denote the estimate $S_t(\widehat{\boldsymbol{\theta}}_g^{(1,1)})$. This means that the deterministic components $g_i(t/T, \boldsymbol{\theta}_{gi})$ have been estimated once, including the intercept $\delta_{i0}$ in (2).

2. Estimate $P_t(\boldsymbol{\theta}_\rho)$ given $\boldsymbol{\theta}_g = \widehat{\boldsymbol{\theta}}_g^{(1,1)}$. This requires a separate iteration because $P_t(\boldsymbol{\theta}_\rho)$ is nonlinear in parameters; see (5) and (6). Denote the estimate $P_t(\widehat{\boldsymbol{\theta}}_\rho^{(1,1)})$.

3. Re-estimate $S_t(\boldsymbol{\theta}_g)$ assuming $P_t(\boldsymbol{\theta}_\rho) = P_t(\widehat{\boldsymbol{\theta}}_\rho^{(1,1)})$. This yields $S_t(\widehat{\boldsymbol{\theta}}_g^{(1,2)})$. Then, re-estimate $P_t(\boldsymbol{\theta}_\rho)$ given $\boldsymbol{\theta}_g = \widehat{\boldsymbol{\theta}}_g^{(1,2)}$. Iterate until convergence. Let the result after $R_1$ iterations be $S_t(\boldsymbol{\theta}_g) = S_t(\widehat{\boldsymbol{\theta}}_g^{(1,R_1)})$ and $P_t(\boldsymbol{\theta}_\rho) = P_t(\widehat{\boldsymbol{\theta}}_\rho^{(1,R_1)})$. The resulting estimates are maximum likelihood ones under the assumption $D_t(\boldsymbol{\theta}_h, \boldsymbol{\theta}_g) = I_N$.

4. Estimate $\boldsymbol{\theta}_h$ from $\boldsymbol{D}_t(\boldsymbol{\theta}_h, \widehat{\boldsymbol{\theta}}_g^{(1,R_1)})$ using $\boldsymbol{P}_t(\boldsymbol{\theta}_\rho) = \boldsymbol{P}_t(\widehat{\boldsymbol{\theta}}_\rho^{(1,R_1)})$. This is a standard multivariate conditional correlation GARCH estimation step as in Bollerslev (1990), because $\boldsymbol{S}_t(\widehat{\boldsymbol{\theta}}_g^{(1,R_1)})$ is fixed and does not affect the maximum and $\boldsymbol{P}_t(\widehat{\boldsymbol{\theta}}_\rho^{(1,R_1)})$ is known. In total, steps 1–4 form the first iteration of the maximisation algorithm. Denote the estimate $\widehat{\boldsymbol{\theta}}_h^{(1)}$.

5. Estimate $\boldsymbol{\theta}_g$ from $\boldsymbol{S}_t(\boldsymbol{\theta}_g)$ keeping $\boldsymbol{D}_t(\widehat{\boldsymbol{\theta}}_h^{(1)}, \widehat{\boldsymbol{\theta}}_g^{(1,R_1)})$ and $\boldsymbol{P}_t(\widehat{\boldsymbol{\theta}}_\rho^{(1,R_1)})$ fixed. This step is analogous to the first part of Step 3. The difference is that $\boldsymbol{D}_t(\widehat{\boldsymbol{\theta}}_h^{(1)}, \widehat{\boldsymbol{\theta}}_g^{(1,R_1)}) \neq \boldsymbol{I}_N$. Denote the estimator $\boldsymbol{S}_t(\widehat{\boldsymbol{\theta}}_g^{(2,1)})$.

6. Estimate $\boldsymbol{P}_t(\boldsymbol{\theta}_\rho)$ given $\boldsymbol{\theta}_g = \widehat{\boldsymbol{\theta}}_g^{(2,1)}$ and $\boldsymbol{\theta}_h = \widehat{\boldsymbol{\theta}}_h^{(1)}$. Denote the estimator $\boldsymbol{P}_t(\widehat{\boldsymbol{\theta}}_\rho^{(2,1)})$. Iterate until convergence, $R_2$ iterations. The result: $\boldsymbol{S}_t(\boldsymbol{\theta}_g) = \boldsymbol{S}_t(\widehat{\boldsymbol{\theta}}_g^{(2,R_2)})$ and $\boldsymbol{P}_t(\boldsymbol{\theta}_\rho) = \boldsymbol{P}_t(\widehat{\boldsymbol{\theta}}_\rho^{(2,R_2)})$.

7. Estimate $\boldsymbol{\theta}_h$ from $\boldsymbol{D}_t(\boldsymbol{\theta}_h, \widehat{\boldsymbol{\theta}}_g^{(2,R_2)})$ using $\boldsymbol{P}_t(\boldsymbol{\theta}_\rho) = \boldsymbol{P}_t(\widehat{\boldsymbol{\theta}}_\rho^{(2,R_2)})$ ($\boldsymbol{S}_t(\widehat{\boldsymbol{\theta}}_g^{(2,R_2)})$ is fixed). The result: $\boldsymbol{\theta}_h = \widehat{\boldsymbol{\theta}}_h^{(2)}$. This completes the second full iteration.

8. Repeat steps 5–7 and iterate until convergence.

For identification reasons, $\delta_{0i}$, $i = 1, \ldots, N$, is frozen to $\delta_{i0} = \widehat{\delta}_{i0}^{(1,R_1)}$. This frees the intercepts in $\boldsymbol{\theta}_{hi}$. Any positive constant would do for $\delta_{i0}$; however, for numerical reasons, the intercepts are fixed to the values they obtain after the first iteration when $\boldsymbol{\theta}_h$ has not yet been estimated a single time.

In practice, in estimating the slope parameters in transition functions it may be useful to apply the transformation $\gamma_{ij} = \exp\{\eta_{ij}\}$, in which case $\gamma_{ij}$ need not be restricted when $\eta_{ij}$ is bounded away from $-\infty$. The motivation for this transformation is that estimating $\eta_{ij}$ instead of $\gamma_{ij}$ is numerically convenient in cases where $\gamma_{ij}$ is large; see Goodwin et al. (2011) or Silvennoinen and Teräsvirta (2016) for discussion.

Another alternative, proposed by Chan and Theoharakis (2011), is to redefine the slope parameter as $\gamma_{ij} = 1/\eta_{ij}^2$ and estimate $\eta_{ij}$. The authors show that this also alleviates the convergence problems sometimes found when $\gamma_{ij}$ is large. Ekner and Nejstgaard (2013) aim at the same effect by rescaling $\gamma_{ij}$ to vary between zero and one.

**Appendix D. Estimated Transition Equations**

This appendix contains the estimated deterministic components in the TV-GARCH Equations (standard deviation estimates in parentheses). Note that the intercept is fixed after the first iteration; hence, it does not have a standard deviation estimate.

ANZ:
$$\widehat{g}_{1t} = 2.28 - \underset{(0.059)}{1.234}(1 + \exp\{-\underset{(1.223)}{5.715}(t/T - \underset{(0.003)}{0.404})\})^{-1}$$
$$+ \underset{(1.518)}{12.316}(1 + \exp\{-\underset{(0.392)}{5.875}(t/T - \underset{(0.002)}{0.571})\})^{-1}$$
$$- \underset{(1.514)}{11.704}(1 + \exp\{-\underset{(0.166)}{4.459}(t/T - \underset{(0.004)}{0.623})\})^{-1}.$$

CBA:
$$\widehat{g}_{2t} = 1.35 - \underset{(0.054)}{0.525}(1 + \exp\{-\underset{(2.545)}{5.638}(t/T - \underset{(0.007)}{0.407})\})^{-1}$$
$$+ \underset{(1.871)}{9.257}(1 + \exp\{-\underset{(0.374)}{5.117}(t/T - \underset{(0.004)}{0.574})\})^{-1}$$
$$- \underset{(1.867)}{8.944}(1 + \exp\{-\underset{(0.252)}{4.504}(t/T - \underset{(0.006)}{0.621})\})^{-1}.$$

$$
\begin{aligned}
\text{NAB:}\qquad \widehat{g}_{3t} =\; &1.07 + \underset{(1.273)}{3.843}\big(1 + \exp\{-\underset{(0.130)}{2.518}(t/T - \underset{(0.034)}{0.303})\}\big)^{-1} \\
&- \underset{(1.114)}{3.491}\big(1 + \exp\{-\underset{(0.329)}{3.787}(t/T - \underset{(0.008)}{0.373})\}\big)^{-1} \\
&+ \underset{(5.692)}{20.026}\big(1 + \exp\{-\underset{(0.229)}{4.926}(t/T - \underset{(0.003)}{0.576})\}\big)^{-1} \\
&- \underset{(5.676)}{20.039}\big(1 + \exp\{-\underset{(0.123)}{4.183}(t/T - \underset{(0.006)}{0.609})\}\big)^{-1}.
\end{aligned}
$$

$$
\begin{aligned}
\text{WBC:}\qquad \widehat{g}_{4t} =\; &2.45 - \underset{(0.554)}{3.120}\big(1 + \exp\{-\underset{(0.124)}{2.194}(t/T - \underset{(0.034)}{0.534})\}\big)^{-1} \\
&+ \underset{(12.524)}{25.782}\big(1 + \exp\{-\underset{(0.158)}{4.569}(t/T - \underset{(0.006)}{0.585})\}\big)^{-1} \\
&- \underset{(12.682)}{23.616}\big(1 + \exp\{-\underset{(0.375)}{4.767}(t/T - \underset{(0.007)}{0.607})\}\big)^{-1}.
\end{aligned}
$$

The locations of the transitions are remarkably similar across transitions. The first transition of the WBC equation is very slow. The effect of the transition extends over the whole estimation period and is the reason for the post-crisis decline in the value of $\widehat{g}_{4t}$; see Figure 5.

## Notes

[1]　　Available also in https://econ.au.dk/research/researchcentres/creates/research/creates-research-papers/supplementary-downloads/rp-2012-09, accessed on 26 January 2023.

[2]　　The operator $vecl(\cdot)$ stacks the subdiagonal elements of its argument matrix.

[3]　　See Explanatory Statement, Banking (prudential standard) Determination 2007, Nos 5, 12 and 15. https://www.legislation.gov.au/Details/F2007L04593/ (accessed on 26 January 2023), https://www.legislation.gov.au/Details/F2007L04600/ (accessed on 26 January 2023) and https://www.legislation.gov.au/Details/F2007L04603/ (accessed on 26 January 2023).

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
