# Peer review of "Building Multivariate Time-Varying Smooth Transition Correlation GARCH Models, with an Application to the Four Largest Australian Banks"

_econometrics, doi:10.3390/econometrics11010005_

Round 1
Reviewer 1 Report
Please see the pdf file.

Reviewer 2 Report
This paper well explain the methodology for building Multivariate Time-Varying STCC–GARCH models, introduced by Silvennoinen and Teräsvirta (in press). In particular, they show all the steps and data-driven decisions required to specify the parametric structure of the model correctly. Finally, the paper is accompanied by an R-package, "mtvgarch." However, I have two small comments:
1) Double check the capital letters on the title.
2)At page 4, you write:
“For the Big Four application we simplify the definition (5) slightly by assuming P_(12)=P_(22)”.
Hence, instead of considering four matrices (P_(11), P_(12), P_(21), P_(22)), you consider P_(1), P_(2), and P_(3). Is this only to have P_t positive definite, or is there also some empirical intuition about it?
